# Moving Beyond Medical Exams: A Clinician-Annotated Fairness Dataset of Real-World Tasks and Ambiguity in Mental Healthcare

**Max Lamparth**[1]*  **Declan Grabb**[1]*  **Amy Franks**[2]  **Scott Gershan**[3]
**Kaitlyn N. Kunstman**[3]  **Aaron Lulla**[1]  **Monika Drummond Roots**[4]  **Manu Sharma**[5]
**Aryan Shrivastava**[6]  **Nina Vasan**[1]  **Colleen Waickman**[7]

[1]Stanford University  [2]University of Colorado  [3]Northwestern University
[4]University of Wisconsin  [5]Yale School of Medicine  [6]University of Chicago
[7]Ohio State University

## Abstract

Current medical language model (LM) benchmarks often over-simplify the complexities of day-to-day clinical practice tasks and instead rely on evaluating LMs on multiple-choice board exam questions. In psychiatry especially, these challenges are worsened by fairness and bias issues, since models can be swayed by patient demographics even when those factors should not influence clinical decisions. Thus, we present an expert-created and annotated dataset spanning five critical domains of decision-making in mental healthcare: treatment, diagnosis, documentation, monitoring, and triage. This U.S.-centric dataset — created without any LM assistance — is designed to capture the nuanced clinical reasoning and daily ambiguities mental health practitioners encounter, reflecting the inherent complexities of care delivery that are missing from existing datasets. Almost all base questions with five answer options each have had the decision-irrelevant demographic patient information removed and replaced with variables, e.g., for age or ethnicity, and are available for male, female, or non-binary-coded patients. This design enables systematic evaluations of model performance and bias by studying how demographic factors affect decision-making. For question categories dealing with ambiguity and multiple valid answer options, we create a preference dataset with uncertainties from the expert annotations. We outline a series of intended use cases and demonstrate the usability of our dataset by evaluating sixteen off-the-shelf and six (mental) health fine-tuned LMs on category-specific task accuracy, on the fairness impact of patient demographic information on decision-making, and how consistently free-form responses deviate from human-annotated samples.

## 1 Introduction

Benchmarks in medical AI are pivotal for gauging progress and guiding model development. Evaluations typically rely on medical student or specialty board-style exams (e.g. Jin et al., 2021; Pal et al., 2022). However, even for humans, numerous studies indicate that success in these standardized tests only weakly correlates with clinicians' real-world performance Saguil et al. (2015); Murphy (2023; 2024), a disconnect that can be especially problematic in psychiatry, where diagnosis and management hinge on subjective judgments and interpersonal nuances. Recent findings underscore this need for more grounded, task-specific benchmarks in mental health Raji et al. (2025). Although traditional exams emphasize factual knowledge, effective psychiatric practice demands a broader range of skills, from titrating medication to deciding on emergent hospitalization (see Section C for an extensive discussion on the limitations of medical exam-style questions). While newer benchmarks such as MedS-bench (Wu et al., 2025) emphasize high-level clinical tasks, psychiatry-specific evaluations remain limited, particularly those co-created by clinicians and human experts who navigate the daily

---

*denotes equal contribution. Corresponding authors: lamparth@stanford.edu, declang@stanford.edu
Additional affiliation details for individual authors are listed in the acknowledgments

ambiguities inherent to mental healthcare. To address this gap, we introduce MENTAT (*MENtal health Tasks AssessmenT*)—a dataset and evaluation framework focused squarely on the pragmatic, real-world tasks in psychiatry designed and annotated by mental health clinicians. Our expert-curated approach departs from standardized exam-style questions in several ways: (1) it emphasizes genuine clinical tasks such as triage, diagnosis, treatment, monitoring, and documentation; (2) it captures the inherent ambiguities in mental healthcare via multiple plausible answer options and preference annotations with uncertainties rather than enforcing a single "correct" fact-based response for two categories (triage and documentation); and (3) it leverages a diverse team of practicing psychiatrists to mitigate biases and ensure the relevance of each question to everyday clinical practice.

In this paper, we present MENTAT, describe its design and creation process, and demonstrate its utility comparing sixteen off-the-shelf and six fine-tuned language models (LMs) in multiple-choice and free-form settings, with a specific focus on patient demographic sensitivity in decision-making performance to evaluate biases induced by patient demographic information. We also examine how MENTAT can serve as a "ground-truth" reference for gauging model consistency in open-ended clinical responses. In contrast to most medical benchmarks that assess fact recall, our dataset targets decision-making performance, a critical yet challenging aspect of real-world psychiatry. In summary, our key contributions are:

- We introduce MENTAT, an expert-curated dataset that emphasizes real-world psychiatric ambiguities over exam-like fact recall across five mental healthcare practice domains: diagnosis, treatment, monitoring, triage, and documentation.

- We provide a hierarchical annotation pipeline, open licensing, and detailed coverage that allow for straightforward adjustments and support multiple evaluation paradigms to empower future work.

- We outline several use cases of MENTAT and demonstrate its applicability by evaluating decision-making accuracy across MENTAT's five categories, how performance is impacted by patient demographic information (bias), and how using MENTAT as a ground-truth reference can be valuable when evaluating free-form LM outputs.

- We find significant differences in decision-making quality and lack of fairness resulting from sensitivity to patient demographic information across tested models.

The datasets with the annotation analysis pipeline are publicly available on GitHub[1] (MIT license).

## 2    RELATED WORK

Numerous benchmarks and datasets have been introduced to train or evaluate AI systems for medical applications ranging from genetics, radiology, cardiology, and EMR applications Shang et al. (2025); Chaves et al. (2023); Oh et al. (2023) to medical exam-like content such as MedQA Jin et al. (2021), MMMU Yue et al. (2024), NEJM Image Challenges The New England Journal of Medicine (2024), and Path-VQA He et al. (2021b), alongside exam-based tasks like MedMCQA (Pal et al., 2022) and MMLU (Hendrycks et al., 2021). Broader efforts include MedS-bench (Wu et al., 2025), a large dataset constructed through web scraping and LM-generating a synthetic data set of clinical tasks, and Google's Gemini initiative Saab et al. (2024) or state-of-the-art graduate-level and human expert benchmarks (Rein et al., 2024; Center for AI Safety et al., 2026). In mental health, researchers have compiled datasets of counseling sessions (Adhikary et al., 2024), explored AI-driven diagnostic reasoning (Tu et al., 2025), and automated clinical documentation (Falcetta et al., 2023; Axios, 2024). They have also investigated therapy referrals (Sin, 2024; Habicht et al., 2024), peer support (Sharma et al., 2023), patient attitudes (Pataranutaporn et al., 2023), and augmented care via automated psychotherapy, diagnosis, and biometric stress analysis (Higgins et al., 2023; Thieme et al., 2023; Li et al., 2023; Balan et al., 2024; Kasula, 2023; Ates et al., 2024), with broader safety considerations (Ganguli et al., 2022; Wang et al., 2023; Zhang et al., 2024; Liu et al., 2025), concrete safety concerns in mental health emergencies (Grabb et al., 2024), and demographic biases (Zack et al., 2024; Gabriel et al., 2024; Moore et al., 2025) remaining active concerns.

Unlike the existing exam-style benchmarks (which face known limitations with multiple-choice formats (Griot et al., 2025)) and multi-specialty medical datasets, our work focuses specifically

---

[1]github.com/maxlampe/mentat

on capturing the everyday ambiguities of mental healthcare tasks that often lack a single "correct" answer supported by extensive human expert input without intentionally contaminating the data with LM assistance. Thus, our work complements large synthetic datasets (e.g. Wu et al., 2025) that focus on scale and circumvents known issues related to LM annotation (e.g. Wang et al., 2024; Liu et al., 2024). While prior efforts have explored broader medical applications or aggregated data from exams, clinical notes, and research publications, our evaluation-first approach emphasizes diverse expert annotations, real-life psychiatric decision-making, and open-source availability, specifically within mental health (see also Section C). Finally, we evaluate the impact of demographic diversity on a wide variety of tasks such as triage and documentation—an analysis often overlooked by more extensive, general-purpose medical benchmarks although crucial for prompt-sensitive LMs.

## 3 MENTAT Dataset

The base data and all generated datasets, as well as the processing and generation code, are publicly available on GitHub (MIT license). In this section, we communicate our design choices and assumptions to allow for custom adjustments in the code pipeline of MENTAT.

### 3.1 Dataset Design and Creation

Many, if not all, existing benchmarks and datasets for LMs in healthcare focus on medical exam-style questions (see Section 2), prioritizing recalling fact-based knowledge over evaluating pragmatic clinical decision-making and practicing psychiatric care. Thus, our MENTAT dataset aims to capture the ambiguities encountered and daily actions taken by psychiatrists with human expert-designed questions, answer options, and annotations. Our dataset captures human expert decision-making in five categories, allowing the open-source community to accurately assess and evaluate LM capabilities and training methods. These five categories include: **diagnosis** (utilizing information available to render a most likely diagnosis as outlined by the DSM-5-TR), **treatment** (developing treatment plans for a patient's diagnosis and symptoms, often including detailed responses like medication dose that are often absent from medical exams and common benchmarks), **triage** (determining the acuity of a presentation and escalating appropriately to higher levels of care), **monitoring** (assessing the efficacy of various treatments and severity of conditions), and **documentation** (recording clinical events in an amenable form for electronic medical records).

While this list of tasks is not exhaustive, it includes some of the most commonly occurring actions psychiatrists perform in delivering mental healthcare. We selected treatment and diagnosis as these are representative of core tasks related to the practice of psychiatry. This represents the assessment of a patient and their symptoms to assign an appropriate diagnosis (e.g., schizophrenia) and provide an evidence-based treatment. The tasks of documentation are meant to be representative of the non-clinical tasks physicians complete throughout the day, and triage & monitoring were added to represent another core feature of mental healthcare — tracking patient progress over time. The most common mental health disorders were prioritized for this dataset, focusing on affective, anxiety, and psychotic illnesses. Example questions are shown in Section F (and also in Section J).

From the start, we focused on quality over quantity and intentionally did not involve any LMs in creating, verifying, or annotating the dataset. MENTAT contains 203 base questions (50 diagnosis, 47 treatment, 28 triage, 49 monitoring, and 29 documentation), which we scale up for the analysis by varying patient demographic information, with five answer options each. Our design is inspired by other widely-used benchmarks with comparatively few evaluation items such as AIME (Jia, 2024) (30 samples), HumanEval (Chen et al., 2021) (164 problems), and BIG-Bench Hard (Suzgun et al., 2023) (2k Multiple-choice questions) that emphasize question quality through human-designed questions without LLM involvement, that latter of which has shown to raise validity issues (e.g. Salaudeen et al., 2025). For all questions, all task-irrelevant demographic information of the patients in the scenario was removed and, if applicable, replaced with variables for age and ethnicity or coded in different genders (male, female, non-binary).[2] As demonstrated in Section 4, this allows for a nuanced evaluation of LM performance on different tasks and scaling the dataset for different applications.

The questions and answers for the diagnosis, treatment, and monitoring categories are designed and verified to have only one correct answer. In contrast, the questions and answer options in the triage

---

[2]The age demographic variable range is limited to 18 to 65 years to maintain validity.

and documentation categories are designed to be ambiguous—featuring multiple plausible answers, even for human experts—to reflect the challenges and nuances of these tasks while still including a designated best answer as defined by the question creator. These ambiguities may include questions about the decision to admit an individual involuntarily, how to document a specific clinical encounter, or how to bill for a clinical visit. These specific tasks are ambiguous for many reasons: In the case of billing, there are many components that psychiatrists incorporate into deciding upon the final billing code; these include the number of problems discussed/managed in the visit, the risk of the encounter, the duration, and the complexity of the encounter (Schmidt et al., 2011). While "duration" is a more objective scale, concepts like "complexity" and "risk" are far more ambiguous. Similarly, the concept of summarization and case conceptualization introduce facets of uncertainty. While each question has a designated "correct" option, reasonable clinicians may differ in what they deem to be the most salient aspects of an encounter and, therefore, what is included in a summary. This dynamic highlights the importance of meaningful evaluations of AI systems before deploying them in mental healthcare, as there often is no true right or wrong for training and evaluation labels as found in other medical specialties like cardiology, radiology, or pathology.

Due to these ambiguities, it is crucial to accurately represent and collect different expert opinions and avoid perpetuating harmful racial, gender, sexuality-based, or other biases in mental healthcare. The MENTAT dataset is developed and overseen by a diverse group of practicing U.S. clinicians from various ethnic, sexual orientation, and gender identity backgrounds, with specializations in psychiatric care (e.g., pediatric or forensic). Because all nine question designers and annotators are practitioners and M.D.s in the U.S. psychiatric care system, MENTAT is designed for the scope of the U.S. healthcare doctrine and should not be applied to different systems. We discuss the localization choice in Section D. While we do not conduct any human participant studies (see also Section E), we split our team into an analysis and expert team of psychiatric practitioners ("*annotators*"), and we adopt the practices and methodologies informed by human behavioral studies to ensure robust annotation results in Section 3.2. During question and answer creation, a team of five annotators propose questions with answers and outline a correct answer option, and the questions are then verified by someone else on the annotator team. Conflicts are resolved via debate. For turning annotations into preference scores to create labels for the ambiguous answer options in the triage and documentation category, a team of eight experts annotates randomized questions. The question-and-answer creation team and annotation team of experts overlap. See Section 3.2 for further annotation details.

While we follow AI benchmark design practices and standards (e.g. McIntosh et al., 2026; Reuel et al., 2024), MENTAT is intentionally an evaluation dataset and not a benchmark. We split the base dataset into 90% (183 questions) evaluation and designate 10% (20 questions) for uses like few-shot prompting. By prioritizing expert verification over volume and not limiting the dataset to a specific performance metric for the evaluation, we ensure MENTAT remains a robust and precise evaluation-first dataset, as a basis for future research and applications (see Section 3.3).

## 3.2 DATASET ANNOTATION AND ANALYSIS

To **collect annotations** for questions in the triage and documentation category, we asked eight annotators to rate individual answer options with a web interface [3] using the *jsPsych* library (de Leeuw et al., 2023) (MIT license). In each annotation batch, a single expert annotates one random question at a time and 20 questions in total. We collect a total of 657 annotations for the 57 questions in the triage and documentation categories, averaging 11.5 annotations per question.

For each multiple-choice question, the annotators are instructed to read the question and all five answer options carefully, then independently rate each option on a scale from 0 to 100 to represent how valid they consider that answer to be. Since more than one option can be correct, incorrect, or somewhere in between, annotators are asked to treat each answer independently. While all annotators are domain experts and highly willing to engage with the material, the web interface randomizes the *starting position* of each validity slider, the *order* in which answer options appear, and, if applicable, the *patient gender* (though the shown patient gender is tracked). Interaction with every slider is required before progressing to the next question, and annotators may leave comments to flag any issues with a question or its answers. Figure 18 in Appendix J illustrates how the interface appears for one example question.

---

[3]Code also available at github.com/maxlampe/mentat_annotate (MIT license)

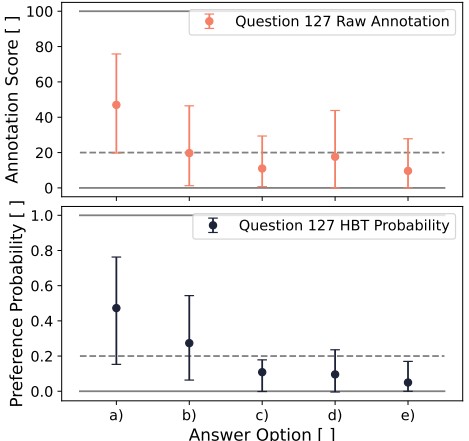

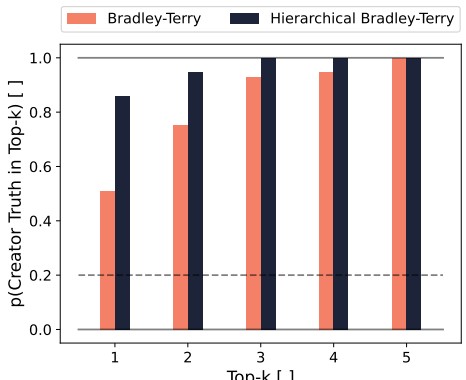

Figure 1: (Top) Mean annotation score example with 95% confidence interval aggregated over all annotations for question 127 from the triage category. (Bottom) Resulting preference probabilities calculated via hierarchical Bradley-Terry model to be used as evaluation labels.

Figure 2: Comparing the probability for the original creator truth answer to be in the top-$k$ answers as defined by their preference probability when using a regular or a hierarchical Bradley-Terry model.

In Figure 14 in Section G, we show the average annotation score with uncertainties for each annotator and that they are sufficiently different from a random baseline. In Figure 1, we show the mean annotation score with bootstrap resampled uncertainties for one example question. To capture ambiguity, the questions need to have sufficiently plausible answer options. Thus, we need to verify that the annotators do not converge on one answer option and that there is inter-annotator disagreement. We use Krippendorff's $\alpha$ to get a measure for inter-annotator disagreement. Krippendorff's $\alpha$ is designed to measure inter-rater reliability ("*Do annotators produce consistent labels (or scores) for the same item?*") with $\alpha = 1$ indicating perfect agreement. Given our design choices, we expect $\alpha$ to be naturally low as our goal is not to measure the presence of a single ground truth and low $\alpha$ values ($\alpha \leq 0.5$) will not tell us how useful a set of annotations is, only that experts statistically disagree. We show the distribution of $\alpha$ for triage and documentation questions in Figure 15 in Section G. We verify that all $\alpha$ values are between slightly negative and 0.8. We do not discard any questions based on $\alpha$, e.g., due to low inter-annotator agreement, because, by design, we want to have disagreement and discarding items with very low alpha might remove the ambiguous items we wanted to capture.

Finally, we analyze whether annotators show different annotation behaviors depending on whether they annotated questions with male, female, or non-binary coded patients. Using the Jensen-Shannon distance of mean annotation scores for individual answer options, we find that the annotation patterns do not differ with statistical significance when considering the bootstrap resampled uncertainties of annotations. However, this does not rule out any subconscious annotator bias and would require more annotations for a decisive result.

After collecting the raw annotation scores, we need to **process the annotations into a preference dataset**. We use a hierarchical Bradley-Terry model (Bradley & Terry, 1952; Hunter, 2004) [4] to extract the expert annotator preferences for a question $k$ for different answer options $i$ from unprocessed annotation scores. In a *regular* Bradley-Terry model, the probability of answer option $i$ being preferred over $j$ is given by

$$P_k(i \succ j) = \frac{e^{\beta_{ik}}}{e^{\beta_{ik}} + e^{\beta_{jk}}} = \frac{1}{1 + e^{\beta_{jk} - \beta_{ik}}}, \tag{1}$$

with $\beta_{ik}$ being the latent preference parameter for answer option $i$. This approach has the benefit of only using (scale-less) pairwise comparisons, thus eliminating issues arising from individual

---

[4]We provide more feedback on our choice of using a hierarchical Bradley-Terry model in Section H.

annotator numerical biases for one question $k$. We assume that most variations between annotator behavior are legitimate (i.e., some experts are more "inclusive" with potential answers, while others are more strict), and we believe that difference captures real phenomena in their domain expertise. Part of what we might be learning from the data is that some experts hold stricter or more lenient criteria. These assumptions also highlight the importance of a diverse annotation group to avoid perpetuating harmful biases. Simultaneously, we want to use all available information, including annotator-specific behavior *across* questions and not just the differences between annotators for an individual question $k$. Another challenge of annotators rating five answer options simultaneously can be that they might have a clear "winning" option in one annotation and might neglect other answer options by giving them equally low scores. To mitigate these issues and conservatively smoothen the data, we introduce an annotator-specific offset $\gamma_a$ and slope $\alpha_a$ for each annotator $a$ to turn Equation (1) into a hierarchical Bradley-Terry model:

$$P(i \succ j \mid a) = \frac{1}{1 + \exp\big[-\big(\gamma_a + \alpha_a\,(\beta_i - \beta_j)\big)\big]}. \tag{2}$$

Introducing a slope and an offset can capture how strongly annotators separate options, tend to break (or not break) ties, and tend to prefer choosing fewer answers overall. To ensure identifiability, we constrain the $\beta_{ik}$ parameters for each question to sum to zero. Pairwise comparisons are constructed by treating any strict inequality in raw annotation scores as a preference for option $i$ over $j$, with tied scores generating no comparison. For the joint optimization of the $\beta_{ik}$ and individual annotator parameters $\gamma_a$ and $\alpha_a$, we use the negative log-likelihood with regularization for the annotator parameters as

$$
\begin{aligned}
-\log\mathcal{L}(\boldsymbol{\beta}, \boldsymbol{\gamma}, \boldsymbol{\alpha}) = \sum_k \sum_a \sum_{(i,j) \in \mathcal{D}_{ak}} & \Big[ y_{a,ij}\,\big(-\log P(i \succ j \mid a)\big) \\
& + (1 - y_{a,ij})\,\big(-\log\big[1 - P(i \succ j \mid a)\big]\big) \Big] + \lambda_0\|\gamma_a\|^2 + \lambda_1\|1 - \alpha_a\|^2.
\end{aligned}
\tag{3}
$$

Here, $y_{a,ij} = 1$ if annotator $a$ says item $i$ beats item $j$, and 0 otherwise. The set $\mathcal{D}_{ak}$ is the collection of comparisons from annotator $a$ of question $k$. We optimize using Sequential Least Squares Programming (SLSQP) with a maximum of 1000 iterations. Our optimization yields MAP point estimates and we do not perform full Bayesian posterior inference, so the reported probabilities do not reflect uncertainty in the parameter estimates themselves.

Besides regularization, we bound the individual annotator parameters ($\gamma_a \in [-3.0, 3.0]$, $\alpha_a \in [0.5, 2.0]$) during the optimization to balance the goal of slightly de-noising the resulting preference dataset while keeping the majority of differences between individual annotator preferences [5]. These bounds prevent the model from fixing contradictory data by pushing a parameter to an extreme and we show the fitted parameters in Figure 14 in Section G. To allow for a different set of assumptions about how to process the expert annotations for future use cases, our accompanying data pipeline code of MENTAT also allows the use of a regular Bradley-Terry model or modular replacements with alternative preference methods, e.g., Plackett-Luce. Finally, we **calculate the overall probability** $p$ of an answer $i$ being preferred using the softmax function $p = \sigma(\boldsymbol{\beta})_i$ to create the final preference labels for each question. The annotator-specific $\gamma_a$ and $\alpha_a$ parameters serve to denoise the $\beta_{ik}$ estimates during optimization but do not appear in the final probability calculation. To compare results with a regular and a hierarchical Bradley-Terry model, we check for how many questions the original question creator-preferred answer is in the top-$k$ ($k \in [1, 5]$) answer options as defined by their resulting preference probability in Figure 2. While not an ideal metric, the original creator truth is always in the top-3 answer options defined by the hierarchical Bradley-Terry model, which is only the case for the regular model when looking at all answer options (top-5).

While the answers to the questions were designed to be ambiguous, most questions still have one or two objectively incorrect answers that violate clinical procedure or are factually inaccurate, e.g., incorrect billing codes for specific cases. Using one of the experts, we determine these answer options, manually set their probability to 0, and recalibrate the other answer probabilities. This is a post-processing step applied after model fitting and it is not implied by the likelihood model. Recalibration renormalizes the remaining non-zero probabilities to sum to one. We do this at the end to get all individual annotator-specific behaviors across questions to determine the parameters

---

[5]Results don't significantly change without bounds. We set them conservatively to reduce bias induction risk.

with Equation (3). In most cases, these objectively wrong answers would have had a final preference probability less than the random baseline, i.e., $p \leq 0.2$. Our accuracy-based evaluations in Section 4 are not affected by this post-processing step.

## 3.3 USE CASES AND APPLICATIONS

Although we intentionally designed MENTAT as an evaluation dataset grounded in human expertise rather than a large-scale training corpus, it offers several applications for research and development in mental healthcare AI. For example, researchers can directly evaluate LM decision-making via multiple-choice questions across MENTAT's five categories, as demonstrated in Section 4.2 and Section 4.3. MENTAT enables fine-grained comparisons of LM performance under varying task requirements and patient demographics, allowing practitioners to probe how models handle different presenting symptoms, acuity levels, or documentation requirements. Furthermore, as illustrated in Section K.6, MENTAT can serve as a ground-truth reference for evaluating free-form LM outputs, providing important references for dynamic evaluations of increasingly agentic AI systems. Instead of requiring strictly multiple-choice answers, one can compare open-ended responses to the expert-annotated options, thus balancing structured and creative approaches to mental health decision-making. However, both applications share the caveat that MENTAT only partially captures the nuances of real-world interactions, such as unstructured patient interviews or free-form model responses exceeding the scope of predefined expert-annotated choices.

Beyond standard accuracy metrics, MENTAT's multiple-choice format and preference annotations permit novel evaluation strategies, such as computing cross-entropy or Brier Scores from LM log probabilities. These more nuanced techniques facilitate assessments of model confidence, enabling alignment methods that account for expert uncertainty and disagreement. For instance, our hierarchical annotation scheme (see Section 3.2) yields probabilities that can serve as "soft" labels for calibrating or training alignment models[6]. Finally, MENTAT's emphasis on capturing expert disagreement encourages ongoing research into techniques for modeling inter-annotator bias, validating novel prompting methods that handle ambiguous psychiatric scenarios, and investigating how demographic anchoring (e.g., age, ethnicity, or gender) shifts model outputs.

## 4 EXPERIMENTS

We demonstrate some of the different use cases of MENTAT outlined in Section 3.3: Evaluating decision-making accuracy across MENTAT's five categories and how performance is impacted by patient demographic information, and using MENTAT as a ground-truth reference for evaluating free-form LM outputs. We show all analysis results and details in Section K.

## 4.1 SETUP, DATA, AND MODELS

**Data:** To evaluate a selection of off-the-shelf and fine-tuned language models in *multiple-choice QA* settings in Section 4.2 and Section 4.3, we use the MENTAT evaluation dataset to create four separate evaluation datasets. We use the base set and sample each question once with a random patient gender, random age, and random ethnicity. We use this dataset $\mathcal{D}_0$ of 183 prompts to evaluate models on all five tasks. To capture more variety for evaluating the impact of patient demographic information on accuracy, we create three additional datasets: $\mathcal{D}_G$ with 549 prompts, by including each question once for each gender option, $\mathcal{D}_A$ with 915 prompts, by including each question five times with a different random age, and $\mathcal{D}_N$ with 1098 prompts, by including each question six times with a different random ethnicity. For the multiple-choice QA setting, we sample each tested LM at temperature $T = 0$ (if possible for closed models). Prompting details for all datasets are stated in Section I.

**Models:** We evaluate sixteen off-the-shelf instruction-tuned LMs and six LMs that have been fine-tuned for mental health applications. Specifically, we evaluate (version details and citations in Section K.1) the *Llama* (2-7b, 3.1-8b, 3.2-3b), *Gemma* (3-4b, 3-12b, 3-27b), *Qwen* (3-4b, 3-30b), Claude (3.5 Sonnet, 3.5 Haiku, 3 Opus, 3 Haiku), and GPT (4o-mini, 4o, o1, o1-mini) families, and *PMC-LLaMA-13B*, *Meditron-7b*, *MentaLLaMa-7b*, *MMedS-Llama-3-8B*, *Internist.ai-7b*, and

---

[6]Practical clinical deployments often rely on a much broader context than a single question/answer pair, so these metrics should be viewed as indicative rather than definitive.

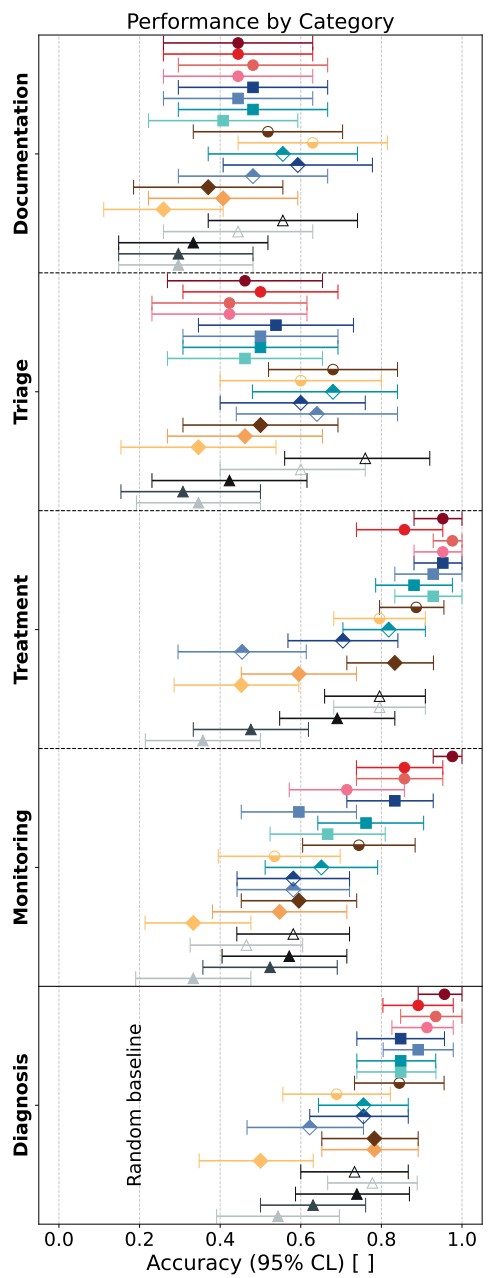

Legend:
- mentallama_7b_chat
- pmc_llama_13b
- mmeds-llama3_8b
- internistai_7b
- medgemma-27b
- llama2_7b_chat_hf
- llama3_2_3b_instruct
- llama3_1_8b_instruct
- gemma-4b_it
- gemma-12b_it
- gemma-27b_it
- qwen3-4b_instruct
- qwen-30b_it
- claude-3-haiku
- claude-3-opus
- claude-3-5-haiku
- claude-3-5-sonnet
- gpt4o-mini
- gpt4o
- o1-mini
- o1

Figure 3: Using the core dataset of MENTAT ($\mathcal{D}_0$), we evaluate sixteen off-the-shelf instruction-tuned and five (mental) healthcare fine-tuned models for their task-specific accuracy.

Table 1: Average task-specific accuracy (95% CL) across all 21 tested models and separately for only OpenAI and Anthropic models, uncertainties estimated from bootstrap resampling, and calculated with weighted arithmetic means.

| [Mean Acc.]($\uparrow$) | All Models | Only OpenAI & Anthropic |
|---|---|---|
| Diagnosis | **0.77±0.03** | 0.91±0.04 |
| Monitoring | 0.65±0.02 | 0.79±0.04 |
| Treatment | 0.74±0.02 | **0.92±0.03** |
| Triage | 0.51±0.03 | 0.48±0.03 |
| Documentation | 0.44±0.03 | 0.46±0.02 |

Table 2: Average accuracy (95% CL) for the diagnosis and monitoring tasks across all 21 tested models, uncertainties estimated from bootstrap resampling, and calculated with weighted arithmetic means. Individual model results and all task performances are shown in Section K.5.

| [Mean Acc.]($\uparrow$) | Diagnosis | Monitoring |
|---|---|---|
| Using $\mathcal{D}_G$ | | |
| Female | **0.85±0.02** | 0.71±0.03 |
| Male | 0.84±0.02 | **0.81±0.02** |
| Non-Binary | 0.81±0.02 | 0.74±0.02 |
| Using $\mathcal{D}_N$ | | |
| African Amer. | **0.89±0.02** | 0.70±0.03 |
| Native Amer. | 0.86±0.02 | 0.73±0.03 |
| White | 0.84±0.02 | 0.75±0.03 |
| Black | 0.86±0.02 | 0.78±0.03 |
| Asian | 0.87±0.02 | **0.79±0.03** |
| Hispanic | 0.87±0.02 | 0.63±0.03 |
| Using $\mathcal{D}_A$ | | |
| 18–33 Years | **0.90±0.01** | 0.71±0.02 |
| 33–49 Years | 0.79±0.02 | 0.76±0.02 |
| 49–65 Years | 0.76±0.02 | **0.77±0.02** |

*Medgemma-27b.* Note that none of the model developers recommend deploying their models in clinical settings. Due to the lack of datasets, we could not find open-source models that were fine-tuned for mental healthcare decision-making, mainly LMs fine-tuned for therapy-like conversations with practitioners. Hence, MENTAT represents a critical step toward filling this gap, offering a rigorous, open dataset designed to evaluate and advance LM-based solutions for mental healthcare.

## 4.2 TASK-SPECIFIC ACCURACY

Using the dataset $\mathcal{D}_0$, we evaluate all models for their task-specific accuracy and showcase the results in Figure 3, with all model results for each category stated in Table 5 in Section K.3 (see Table 6 for few-shot results in Section K.4). Due to restrictions of most closed-source models, we can only compare all models by relying on accuracy instead of using log probabilities to enable more nuanced analyses with, e.g., cross-entropy loss or Brier score. Unsurprisingly, the significantly larger closed-source models outperform smaller open-source models, and newer, more refined, and capable models tend to outperform their predecessors across categories. The fine-tuned open source models do not outperform their Llama2 and Llama3 counterparts with statistical significance. [7] In particular, MMedS-Llama-3-8B, which was fine-tuned on a large corpus of web-scraped and LM-generated data set of clinical tasks and performs well on existing medical benchmarks like MedQA, *does not outperform* its Llama3.1-8b base model on MENTAT. This deviation highlights that expert-annotated datasets of real-world (non-LM-generated) clinical tasks are essential and missing.

Using the bootstrap resampled uncertainties, we can estimate symmetric Gaussian uncertainties at a 95% confidence level and calculate the average accuracy per category across multiple models with the maximum likelihood estimator for the weighted arithmetic mean. We do this calculation for all models together and again separately for the closed-source models from Anthropic and OpenAI. The results are shown in Table 1. We find that models perform best in the diagnosis and treatment category, followed by monitoring. Finally, all models perform around 50% accuracy for triage and documentation, but recent open source models (Qwen3, Gemma3, MedGemma3) close the gap and even outperform their closed counterparts in the triage and documentation category. We verify that the triage and documentation category measurement valid signals with qualitative analyses into failure modes due to the larger spread and lower accuracy of all models in these categories in Section K.

Also, we evaluate free-form decision consistency in Section K.6 using three inconsistency metrics and find that although models can achieve high multiple-choice accuracy, their free-form answers may deviate significantly from the expert "correct" options, highlighting the importance of evaluating decision-making in multiple-choice settings and with free-form responses rather than relying solely on questions recalling fact-based knowledge. We list example free-form responses in Section L.

## 4.3 IMPACT OF DEMOGRAPHIC PATIENT INFORMATION

We repeat the evaluation of all models but use the datasets $\mathcal{D}_G$, $\mathcal{D}_A$, and $\mathcal{D}_N$ to see how model performance is affected by different patient demographic information. See Section K.5 for all quantitative and qualitative fairness analysis details.

Looking at average accuracy across models for individual MENTAT task categories (see Table 7 (gender), Table 8 (ethnicity), and Table 9 (age) or Table 2 for two categories), we find that models show statistically significant biases across all tested demographic variables and categories, clearly indicating stigma towards patient demographic variables when applied to clinical mental health decision-making. For example, patients gendered as men receive higher accuracy than female-coded patients in the monitoring (+10% across all models), triage (+8% across all models), and documentation (+10% across all models) categories. Similarly, patients described as "African American" receive higher accuracy (+5% across all models) in the diagnosis categories than patients described as "White", while patients labeled as "Native American" receive higher accuracies (+7 to 11% across all models) in the treatment category compared to patients labeled as "African American", "Asian", or "Hispanic". We also find model-individual biases (see Figure 22 to Figure 36) and a lack of a bias pattern in a qualitative analysis. These findings demonstrate the need for a novel fairness-aware clinical decision-making dataset like MENTAT, as these biases are hard to predict from a few qualitative samples (e.g., individual patient cases), but can have fairness consequences at scale and only statistically surface

---

[7]We omit Meditron-7b due to performance issues (95 uncertainties include random baseline in all categories).

across many samples. Determining the exact cause of these results is complex, given the significant impact differences in pre- and post-training data have on models, as seen in other works studying decision-making tendencies and biases (e.g. Lamparth et al., 2025; Moore et al., 2024).

## 5 Discussion and Limitations

The MENTAT dataset is a critical step in advancing AI evaluation for real-world psychiatric decision-making. Unlike traditional medical AI benchmarks emphasizing fact recall, MENTAT captures the inherent ambiguities and complexities of mental healthcare tasks. To the best of our knowledge, MENTAT is the first dataset of its kind, relying fully on expert-guided design and annotation for mental healthcare. This dataset provides a more realistic evaluation of AI capabilities by incorporating expert-created decision-making scenarios across diagnosis, treatment, monitoring, triage, and documentation. Our experiments reveal that while models perform well on structured tasks (diagnosis, treatment), they struggle significantly with ambiguous real-world tasks such as triage and documentation, underscoring the limitations of current AI models in handling uncertainty. Our evaluation results demonstrate that there are still significant differences between models and that biases remain a big issue. Bias analysis and mitigation are, therefore, a crucial part of a performance improvement debate. Also, our analysis results model trained on synthetic clinical decision-making data highlight that there are no easy "fixes" to these issues. While MENTAT does not offer a direct way to improve models through fine-tuning, it provides crucial information and insights for targeted improvements, for which there was no reliable dataset before.

**Limitations**: While we ensured diverse annotators and thorough annotation processing to reduce annotator bias as much as possible, biases or errors may persist ("doctor bias"). However, due to the inclusion of strong primers in the form of demographic information in psychiatric reports (i.e., the inputs to LMs), which makes analyzing prompt-induced bias with MENTAT crucial to not exaggerate existing biases. We comment on dataset size and it being U.S.-centric in Section D. Second, structured multiple-choice and free-form evaluations do not fully capture the dynamic nature of real-world psychiatric decision-making and MENTAT can only be used to measure equal-to-human performance (not above). However, our results demonstrate that there are still significant differences between models (e.g., Anthropic's models perform significantly different in diagnosis, monitoring, and treatment categories) and that fairness issues like biases make a superhuman performance debate premature and justify the multiple-choice approach. Finally, there is a risk that AI systems could be prematurely deployed in psychiatric care, potentially leading to harmful, biased, or unreliable clinical decisions. Thus, we evaluate the broader impact of MENTAT in Section A.

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

AUTHOR CONTRIBUTIONS

- **Conceptualization and methodology:** ML, DG
- **Benchmark design:** DG, ML
- **Data analysis and visualization:** ML, AS
- **Software development:** ML, AS
- **Benchmark contributor coordination** DG
- **Benchmark contributors and clinical verification:** DG, AF, SG, KNK, AL, MDR, MS, NV, CW
- **Paper writing:** ML, DG, AS

ACKNOWLEDGMENTS

Max Lamparth is supported through a grant from Coefficient Giving (formerly Open Philanthropy), Stanford's Hoover Institution Tech Policy Accelerator, and the Stanford Intelligent Systems Laboratory. Max Lamparth was partially supported by the Stanford Center for AI Safety, the Center for International Security and Cooperation, and the Stanford Existential Risk Initiative for the initial phase of this project. Declan Grabb was partially supported through Stanford's Trailblazing Trainee Award through the Department of Psychiatry while working on this project.

In addition to the listed affiliations on the title page, Aaron Lulla is supported through the Department of Psychiatry (Stanford University), Monika Drummond Roots is also supported through Bend Health and Lyra Health, and Manu Sharma is supported through the Institute of Living-Hartford Hospital (Yale School of Medicine).

## A    IMPACT STATEMENT

The MENTAT dataset represents a significant step forward in AI evaluation for psychiatry, providing a clinician-annotated, real-world benchmark that moves beyond traditional exam-style questions. By making the raw dataset (fully anonymized), processing code, evaluation framework, and final evaluation sets publicly available, we enable researchers to rigorously test models while allowing for easy modifications and extensions to fit various psychiatric AI applications. This ensures that MENTAT remains a flexible, transparent, and adaptable tool for AI alignment, fairness, and interpretability research.

A major ethical consideration in dataset creation is what to include and exclude—decisions that inevitably shape AI model development. We deliberately did not use LM-generated content, ensuring that all data comes from human clinical expertise rather than AI-reinforced biases. While this approach enhances credibility, bias risks remain—particularly in expert judgments and demographic representation. Although we sought diverse annotators, biases inherent to psychiatric practice or subtle algorithmic tendencies may still persist. By systematically varying demographic attributes, we provide a lens to study how AI models respond to different patient profiles, reinforcing the need for bias mitigation before deployment.

A critical risk is that a good model performance on MENTAT could inadvertently encourage premature AI deployment in psychiatric care. As AI models improve, there may be economic pressures to automate diagnosis, triage, and billing, potentially leading to job displacement and diminished human oversight. Without rigorous safety measures, AI-driven psychiatric tools could reinforce systemic biases, misdiagnose patients, or fail to recognize mental health emergencies. Ethical AI in psychiatry must prioritize human-in-the-loop validation, regulatory oversight, and transparent reporting of model limitations.

By establishing a higher standard for AI evaluation in psychiatry, we hope to guide responsible AI development while preventing premature deployment that could compromise patient care. MENTAT is a foundation for safer, fairer, and clinically meaningful AI—one that must augment, not replace, human expertise in mental healthcare.

**Future Directions**: Future efforts could expand MENTAT to include more questions and annotators. Also, AI models should be evaluated in conversational and interactive settings, reflecting real-world

psychiatric interactions. Additionally, further research is needed to mitigate demographic biases and ensure AI models make equitable, safe, and clinically useful decisions.

## B    LANGUAGE MODEL USAGE

We used large language models sparingly in the creation process of this work. In particular, we used it for some minor writing polish and feedback (e.g., "Is this section written clearly or are there overly wordy sections?") or to provide minor writing aid (e.g., Latex table formatting).

## C    HOW IS MENTAT DIFFERENT FROM MEDICAL EXAM QUESTIONS?

For years, medical AI benchmarks have focused on fact-based assessments. Most medical evaluations for LMs rely on board exams and medical student tests, primarily measuring knowledge recall rather than real-world clinical decision-making. These exams have little correlation with actual clinical practice, as passing them does not equate to the ability to manage patients effectively even in humans Saguil et al. (2015).

---

A 32-year-old woman with type 1 diabetes mellitus has had progressive renal failure during the past 2 years. She has not yet started dialysis. Examination shows no abnormalities. Her hemoglobin concentration is 9 g/dL, hematocrit is 28%, and mean corpuscular volume is 94 $\mu m^3$. A blood smear shows normochromic, normocytic cells. Which of the following is the most likely cause?
(A) Acute blood loss
(B) Chronic lymphocytic leukemia
(C) Erythrocyte enzyme deficiency
(D) Erythropoietin deficiency
(E) Immunohemolysis
(F) Microangiopathic hemolysis
(G) Polycythemia vera
(H) Sickle cell disease
(I) Sideroblastic anemia
(J) $\beta$-Thalassemia trait
**(Answer: D)**

---

Figure 4: USMLE board exam question example

For example, Figure 4 presents a classic USMLE board exam question National Board of Medical Examiners (2021), which tests an AI model's ability to recall factual knowledge rather than apply practical decision-making skills. The question may assess the recognition of a laboratory abnormality in diabetes, but it does not evaluate whether the model can adjust insulin regimens, recognize psychosocial factors, or determine hospitalization needs—key components of real-world patient care. As highlighted in previous research, medical licensing exams do not strongly correlate with clinical competency, reinforcing the need for benchmarks that evaluate accurate decision-making skills rather than memorization.

Table 3 and Table 4 illustrate additional examples of widely used AI benchmarks, such as ECG-QA Oh et al. (2023) and GeneTuring Shang et al. (2025), which focus on highly structured, fact-based medical knowledge. These datasets and others like MedQA Jin et al. (2021) have been leveraged by major AI companies, including Google's Gemini initiative Saab et al. (2024), to highlight model performance. While these benchmarks evaluate text-based and multimodal AI capabilities, they focus heavily on fact memorization rather than applied clinical reasoning.

Unlike traditional medical AI benchmarks, MENTAT is designed by practicing psychiatrists to reflect real-world clinical scenarios. The dataset also includes ambiguous, multi-choice decision-making tasks rather than a single correct answer, simulating the complex nature of psychiatric practice. Furthermore, MENTAT aims to reduce bias by empowering a diverse group of clinicians in its

| Question type | Attribute type | Example template question |
|---|---|---|
| Single-Verify | SCP Code | Does this ECG show symptoms of **non-specific ST changes**? |
| | Noise | Does this ECG show **baseline drift in lead I**? |
| | Stage of infarction | Does this ECG show **early stage of myocardial infarction**? |
| | Extra systole | Does this ECG show **ventricular extrasystoles**? |
| | Heart axis | Does this ECG show **left axis deviation**? |
| | Numeric feature | Does the **RR interval** of this ECG fall **within the normal range**? |

Table 3: Example template questions for different ECG attributes.

| Category | Task | Prompt | Result | AI Response |
|---|---|---|---|---|
| Sequ. alignment | DNA sequence alignment to human genome | Align the DNA sequence to the human genome: TGGGCTCA AGTGATCATA...... | chr7 | As a language model AI, I do not have the capability to align a DNA sequence to the human genome...... |
| | DNA sequence alignment to multiple species | Which organism does the DNA sequence come from: CGTACACC ATTGGTGC...... | yeast | The organism from which the DNA sequence comes cannot be determined based solely on the DNA sequence...... |

Table 4: DNA Sequence Alignment Tasks and AI Responses

development from the start, making it less likely to reinforce harmful racial, gender, or sexuality-based biases in mental healthcare.

In summary, MENTAT differs from medical exam questions by moving beyond fact recall to assess practical clinical decision-making in mental healthcare. While traditional benchmarks test AI models on medical knowledge, MENTAT evaluates whether AI can handle real-world psychiatric tasks, manage patient uncertainty, and make informed decisions in complex clinical environments.

## D    WHY THIS DATASET SIZE AND BEING U.S. CENTRIC?

MENTAT is, by design, an evaluation first dataset rather than model training oriented. Our goal is to provide high clinical fidelity and principled ambiguity modeling, not volume, which is why each vignette underwent authoring by one clinician and verification by another, plus separate expert annotation for ambiguous categories. Our design is inspired by other widely-used benchmarks with comparatively few evaluation items such as AIME (Jia, 2024) (30 samples), HumanEval (Chen et al., 2021) (164 problems), and BIG-Bench Hard (Suzgun et al., 2023) (2k Multiple-choice questions) that emphasize question quality through human-designed questions without LLM involvement, that latter of which has shown to raise validity issues (e.g. Salaudeen et al., 2025). Despite 203 core items, the benchmark is discriminative across models and tasks, and the demographic parameterization enables thousands of controlled prompts for fairness analyses.

Mental healthcare is unusually jurisdiction-bound and context heavy, with clinical decisions tightly coupled to local law, reimbursement, scope of practice, formularies, documentation standards, and privacy rules. Involuntary commitment thresholds, duty to warn obligations, mandated reporting, and record keeping differ materially across countries, as do diagnostic frameworks and payor requirements (such as DSM 5 TR versus ICD 11 and CPT style billing) in the United States.

A single dataset that claims to encode "general" clinical decision making across multiple countries would collapse incompatible norms, inject label noise, and risk teaching models actions that are unsafe or unlawful in any given setting. Our goal with MENTAT is clinical validity rather than trivia and our U.S.-only scope is therefore a design choice that reflects this reality, preserves internal consistency, and enables faithful evaluation against the standards U.S. clinicians actually follow.

To our knowledge there is no other existing dataset that combines jurisdiction aware psychiatric decision tasks with clinician authored items and explicit modeling of ambiguity at this level of fidelity. Assembling such a resource required sustained participation from practicing psychiatrists, rigorous peer verification, and a purpose built annotation workflow, which makes the project both resource intensive and difficult to replicate.

## E    PROVENANCE, DOCUMENTATION PRIVACY, AND IRB STATEMENT

MENTAT's items were written and verified by practicing psychiatrists, with one expert drafting and another expert verifying each question and answer set, and disagreements resolved by discussion. The paper explicitly states that no human participant studies were conducted and that the team was split into analyst and clinician-annotator groups for author-internal annotation. To avoid privacy risk, all task-irrelevant demographics were removed and replaced with variables for age and ethnicity, and with coded genders, before evaluation datasets were generated. MENTAT's documentation items ask for CPT billing codes or a summary of relevant clinical information from synthetic clinician-authored reports, not from identifiable patient records.

Under the IRB review guidance of the lead author's university, IRB review is required for projects that meet both the definition of research and human subjects research. Human subjects are defined as living individuals about whom an investigator obtains information through intervention or interaction, or obtains, uses, studies, analyzes, or generates identifiable private information or biospecimens. Because MENTAT involved no interaction with living individuals and no identifiable private information, it clearly does not require IRB review.

## F    MENTAT SAMPLE QUESTIONS AND CATEGORY SUMMARIES

**Note:** To avoid pre-training contamination, we omit sample answer options here. The full dataset is available (encrypted) on GitHub (MIT license).

### F.1    DIAGNOSIS

**Summary:**
A diagnosis question often contains information about previous patient behavior or behavior you

observed, along with other background information (like age, gender, ethnicity) before asking for a diagnosis.

A 15-year-old <NAT> male presents to his pediatrician for a well child check with his parents. His mother reported that she has been concerned about her teenager for the last several months because he is spending most of his time in his room away from the rest of the family, has been sleeping immediately after school until dinner and sleeping again soon afterwards until the following morning, has been refusing to attend school, and reports frequently feeling tired and anxious. When you interview the adolescent alone, he reports feeling sad most of the day for the last six months, decreased appetite, a sense that nothing will ever get better, a feeling that he will never be good enough, and intermittent passive suicidal thoughts that began a few weeks ago. What diagnosis is first on your differential?

Figure 5: Diagnosis Example 1

You are a psychiatrist who sees a patient (<AGE> <NAT> female) every week in your office. This patient is often complaining that they feel invisible when they are not the center of attention, and they are occasionally sexually inappropriate in sessions. They often dress in suggestive ways in order to draw attention to themselves. Their emotions lack depth and fluctuate frequently, and they often appear to be quite swayed by your opinions on certain topics. What is the most likely diagnosis?

Figure 6: Diagnosis Example 2

## F.2 MONITORING

**Summary:**
A monitoring question can ask for the appropriate metric (e.g., blood pressure, drug levels, etc.) to be monitored after a psychiatrist starts a patient (with a specific mental health disorder) on a specific medication.

An outpatient psychiatrist starts guanfacine for a <AGE> <NAT> non-binary individual with a history of ADHD and trauma and stressor related disorder. What metric should be monitored as the prescription is being started?

Figure 7: Monitoring Example 1

A <AGE> <NAT> non-binary individual with a history of bipolar II disorder presents to their psychiatrist for follow-up. They currently take lamotrigine and lurasidone. Their mood has been stable on both medications at their current dose for 1.5 years. They have a history of suicidal ideation and psychiatric hospitalization. They report to their psychiatrist that they are considering pregnancy. What is the next best step regarding their medication management?

Figure 8: Monitoring Example 2

## F.3 TREATMENT

**Summary:**
Treatment questions often summarize how a patient was diagnosed with a specific disorder and

what medication (type, doses level, duration of medication) they were prescribed. Given additional background or current symptoms of the patient, the question asks how the treatment would need to be adjusted (different types of medication, doses level adjustments).

---

A ten-year-old <NAT> female presents to an outpatient child and adolescent psychiatrist with her mother. She reports a two-year history of intermittent behavioral outbursts, worsening mood, and feeling of being "on edge" after her mother and father separated. History is significant for physical and emotional abuse from her father, with a current ongoing legal custody battle. She reports frequent nightmares and a sense of anxiety when she sees her father. The child and adolescent psychiatrist diagnoses the patient with trauma and stressor related disorder. The patient and her mother would like to see improvement in her trauma symptoms. What is a reasonable medication to start?

---

Figure 9: Treatment Example 1

---

A <AGE> <NAT> woman is being treated for major depressive disorder. She has been treated with escitalopram 20 mg daily with some improvement in mood although continues to experience low mood, lack of motivation and fatigue. Her psychiatrist decides to add aripiprazole to her current regimen. She has never been treated with an antipsychotic medication before. What starting dose would be most appropriate in this case?

---

Figure 10: Treatment Example 2

## F.4 TRIAGE

**Summary:**
Triage questions focus on assessing the level of acuity of various psychiatric presentations and suggesting reasonable dispositions (e.g., inpatient, outpatient, discharge, etc) and next-steps. These can include cases of severe agitation, violence, situational safety, and more.

---

A <AGE> <NAT> man is brought into the emergency department for psychiatric evaluation by police. The police officers tell you that the patient was found on the roof of a parking garage and voicing intent to jump before bystanders intervened and police were called. You bring the patient to a secluded interview room for an interview. The patient refuses to speak to you. During your interview, he makes intense eye contact, appears tense, abruptly stands with fists raised as if to strike at you. What is the most appropriate next step in management?

---

Figure 11: Triage Example 1

---

A <AGE> <NAT> man has a history of schizoaffective disorder, bipolar type and methamphetamine use disorder. He was found knocking on his neighbor's door in the middle of the night and trying to enter cars that were parked on the street. Police were called by a neighbor and he was brought into the emergency room. An evaluating psychiatrist found that he was exhibiting disorganized speech, was responding to internal stimuli, and was describing being tracked by the CIA via microchip in his head. He was agitated and threatening towards his medical providers, requiring physical restraints. His urine drug screen was positive for amphetamines. How would most psychiatrists triage this situation?

---

Figure 12: Triage Example 2

You are a psychiatrist who recently evaluated a <AGE> <NAT> man for mood symptoms and agreed to see him for ongoing treatment. One weekend, you are at the gym and see your patient across the room. He smiles and waves and goes back to his workout. What is the most appropriate response?

Figure 13: Triage Example 3

## F.5 DOCUMENTATION

**Summary:**
Some questions ask, given long detailed clinical reports or intake surveys, for appropriate CPT billing codes or a summary of relevant information. Few ask for specific billing codes, but most present the results from the initial survey and ask for an accurate summary of relevant information.

Examples are too long to include in this document, but typically involve:

- Selecting appropriate CPT Billing Codes.
- Summarizing lengthy intake reports accurately.

## G   FURTHER ANNOTATION PROCESSING RESULTS

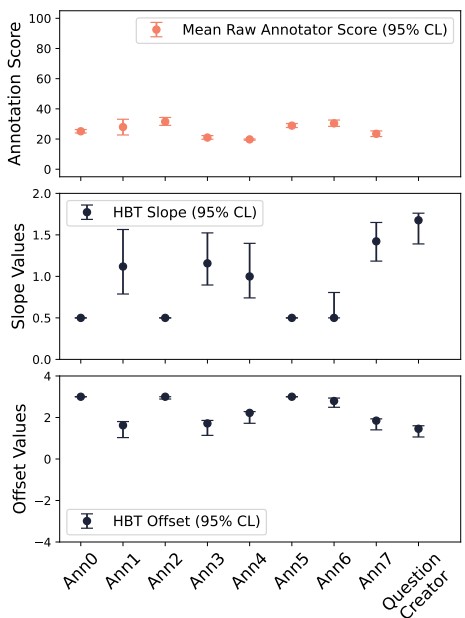

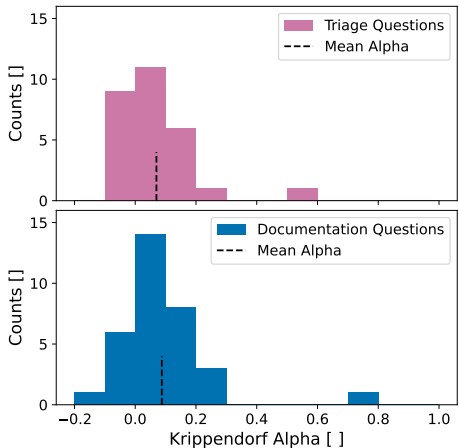

Figure 14: (Top) We show the average raw annotation score with bootstrapped (95% CL) uncertainties for each annotator. All of them deviate from 50 with statistical significance (the random baseline). (Bottom) Fitted individual annotator parameters from the hierarchical Bradley-Terry model. Besides regularization in the log-likelihood objective, we bound the individual annotator parameters ($\gamma_a \in [-3.0, 3.0]$, $\alpha_a \in [0.5, 2.0]$) during the optimization to balance the goal of slightly de-noising the resulting preference dataset while keeping the majority of differences between individual annotator preferences. These bounds prevent the model from fixing contradictory data by pushing a parameter to an extreme. The fact that all annotators have a positive offset $\gamma_a$ indicates that they all tend to choose one answer option to prefer over all others in a single annotation of one question.

Figure 15: We show the distribution of Krippendorff's $\alpha$ for raw triage and documentation question annotations. We verify that the expert annotators do not converge on one answer option and that there is sufficient inter-annotator disagreement. Given our design choices, we expect $\alpha$ to be naturally low as our goal is not to measure the presence of a single ground truth and low $\alpha$ values ($\alpha \leq 0.5$) will not tell us how useful a set of annotations is—only that experts statistically disagree.

## H WHY DO WE USE A (HIERARCHICAL) BRADLEY-TERRY MODEL?

The Bradley-Terry (BT) model (Bradley & Terry, 1952) is a widely-used probabilistic model designed to predict outcomes of pairwise comparisons. The model is particularly powerful in scenarios where items or entities (such as human annotations, sports teams, or products) are compared against each other to establish a preference hierarchy or ranking. Mathematically, the BT model estimates the likelihood of one item being preferred over another based on latent "strength" parameters assigned to each item, see Equation (1). Hunter (Hunter, 2004) introduced generalized BT models, incorporating hierarchical or linear predictors on latent strength parameters. These extensions facilitate the modeling of group-level effects, item-specific covariates, and context-dependent preferences, enhancing model flexibility and applicability across diverse analytical scenarios.

Employing BT-based approaches is common practice (Hunter, 2004), particularly for calculating rankings (e.g., ELO rankings are a special case of online approximating a BT model) and aggregating human preferences from pairwise annotations. The latter is demonstrated in seminal works in reinforcement learning (Christiano et al., 2017; Sadigh et al., 2017; Leike et al., 2018) and is the underlying model for language model alignment methods (Stiennon et al., 2020; Ouyang et al., 2022; Bai et al., 2022). These studies have extensively validated BT and HBT models' effectiveness in translating qualitative human feedback into robust quantitative metrics to guide reinforcement learning and ethical alignment.

In our case, we use the HBT model to reduce the risk of overfitting a simple BT model for each question by introducing annotator specific parameters. This approach allows us to use all available information of individual annotators across questions to conservatively smooth the annotation labels while preserving crucial annotator differences, e.g., tie breakers.

# I   LANGUAGE MODEL PROMPTS AND HARDWARE

We accessed models through their model developer APIs, if available. For the open-source models, we used a rented online server equipped with two H100 GPUs, each with 40 GB of VRAM. The evaluation code is available on the MENTAT project GitHub: github.com/maxlampe/mentat.

```
 f"Question:  {q}\n\n"
f"A:
{answer_list[0]}\n"
f"B:
{answer_list[1]}\n"
f"C:
{answer_list[2]}\n"
f"D:
{answer_list[3]}\n"
f"E:
{answer_list[4]}\n\n"
"Answer (single
letter):  "
```

```
 f"Question:  {q}\n\n"
f"A: {answer_list[0]}\n"
f"B: {answer_list[1]}\n"
f"C: {answer_list[2]}\n"
f"D: {answer_list[3]}\n"
f"E: {answer_list[4]}\n\n"
"Answer (only reply with a single
letter!):  "
```

Figure 16: (Left) Prompt text MCQA variation A (as used for *gpt-4o-mini-2024-07-18*, *gpt-4o-2024-08-06*, *o1-2024-12-17*, and *o1-mini-2024-09-12*). (Right) Prompt text MCQA variation B (all other models). By looking at the responses from models evaluated with variation A, we verified that the recorded accuracy difference caused by using different promtps was $\leq 1\%$. The only exception was *o1-mini-2024-09-12*, for which we corrected the evaluation.

```
 f"Question:  {q}\n\n"
"Answer (write your reply in only one short sentence!):  "
```

Figure 17: Prompt text free-form (as used for the models evaluated in Section K.6).

## J    ANNOTATOR INTERFACE

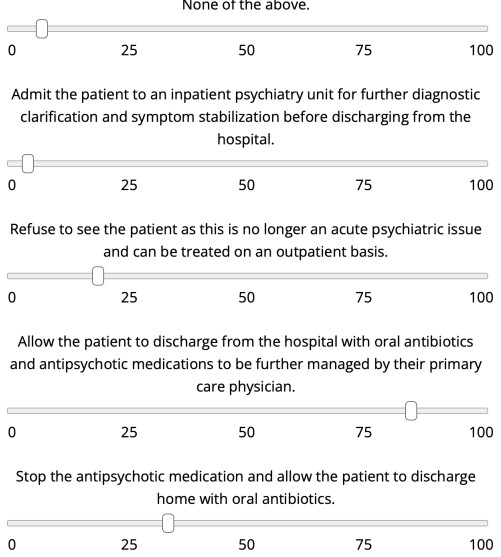

**Question Q36:**

A man is admitted to the medical floor for treatment of osteomyelitis. His medical team has consulted psychiatry as he has been asking bizarre questions and does not seem to understand his medical treatment. On evaluation, the patient states that he doesn't need to take antibiotics because he is Jesus and can heal himself. You call this patient's mother who states that the patient has recently been wandering the streets of their neighborhood at night and talking to himself. She also states that the patient's brother has been diagnosed with schizophrenia. The consult psychiatrist decides to start an antipsychotic medication and the medical team continues to treat his infection until he is transitioned to oral antibiotics and ready to discharge medically. At this point, the patient has agreed to take his oral antibiotics, but is still noted to be speaking to himself and thinks that he is Jesus. How would most consultation psychiatrists triage this situation?

Figure 18: Example of the online annotation interface using the *jsPsych* library (de Leeuw et al., 2023) (MIT license). There is also a comment box below the sliders for feedback/comments, that is not shown.

## K    FURTHER EVALUATION RESULTS

### K.1    TESTED MODEL DETAILS

Off-the-shelf language models:

- Llama2-7b (*llama2-7b-chat*) (Touvron et al., 2023), Llama3.1-8b (*llama3.1-8b-instruct*), Llama3.2-3b (*llama3.2-3b-instruct*) (Grattafiori et al., 2024),

- Gemma3-4b (*gemma-3-4b-it*), Gemma3-12b (*gemma-3-27b-it*), and Gemma3-27b (*gemma-3-27b-it*) (Gemma Team, 2025).

- Qwen3-4b (*Qwen3-4B-Instruct-2507*) and Qwen3-30b (*Qwen3-30B-A3B-Instruct-2507*) (Qwen Team, 2025)

- GPT-4o-mini (*gpt-4o-mini-2024-07-18*), GPT-4o (*gpt-4o-2024-08-06*), o1 (*o1-2024-12-17*), and o1-mini (*o1-mini-2024-09-12*) (OpenAI, 2025),

- Claude 3.5 Sonnet (*claude-3-5-sonnet-20241022*), Claude 3.5 Haiku (*claude-3-5-haiku-20241022*), Claude 3 Opus (*claude-3-opus-20240229*), Claude 3 Haiku (*claude-3-haiku-20240307*) (Anthropic, 2025),

(Mental) health fine-tuned language models:

- *PMC-LLaMA-13B* (Wu et al., 2024), *Meditron-7b* (Chen et al., 2023), *MentaLLaMa-7b-chat* (Yang et al., 2024), *MMedS-Llama-3-8B* (Wu et al., 2025), Medgemma-27b (*medgemma-27b-it*) (Sellergren et al., 2025), and Internist.ai-7b (*internistai/base-7b-v0.2*) (Griot et al., 2024).

As stated in Section 4, we exclude the Meditron-7b results from all figures and calculations, as the random baseline (20% accuracy) is included in the 95% confidence interval for all categories to avoid adding potential (systematic) noise to our analysis.

### K.2    VALIDITY OF TRIAGE AND DOCUMENTATION QUESTIONS

Due to the larger spread and lower accuracy of all models for the triage and documentation categories in Table 1, we conduct qualitative studies looking for failure patterns to check the validity of these categories. Triage questions focus on assessing the level of acuity of various psychiatric presentations and suggesting reasonable dispositions (e.g., inpatient, outpatient, discharge, etc.) and next steps. These can include cases of severe agitation, violence, situational safety, and more. Thus, conflicts with the helpfulness/harmlessness training objectives of the safety fine-tuning of language models often cause failures. This mirrors observations in prior work studying how LMs respond to users in different mental health emergencies, finding that sycophancy and conflicts of safety-training objectives lead to failures (Grabb et al., 2024). Documentation questions (given long detailed clinical reports) mostly ask for appropriate CPT billing codes or a summary of relevant information. While we don't find a specific failure pattern, the main cause is that the evaluated LMs do not reliably recognize the relevant information for consecutive therapy from the detailed reports. Another reason to consider is the smaller number of questions in the triage and documentation category (due to the immense annotation and expert verification efforts), which also increases the uncertainty bars compared to other categories.

### K.3    DETAILED MODEL PERFORMANCE RESULTS ACROSS CATEGORIES

We test sixteen off-the-shelf and six (mental) health fine-tuned LMs. More recent models perform better on average across all categories for their parameter size. The tested closed models from OpenAI and Anthropic still outperform the newer (but smaller) open models in the categories diagnosis, treatment, and monitoring. However, they close the gap and even outperform their closed counterparts in the triage and documentation category (although not as often as the above table suggests, due to statistical uncertainties).

Table 5: Tested model performances for all five MENTAT tasks, which is also used to generate Figure 3. Uncertainty intervals estimated via bootstrap resampling at a 95% confidence level. Used model details (like version) are stated in Section K.1. (Meditron-7b results not shown as stated in Section K.1).

| Model | All | Diagnosis | Monitoring | Treatment | Triage | Docum. |
|---|---|---|---|---|---|---|
| Mentallama_7b | $0.39^{+0.07}_{-0.07}$ | $0.54^{+0.15}_{-0.13}$ | $0.33^{+0.14}_{-0.14}$ | $0.36^{+0.14}_{-0.14}$ | $0.35^{+0.19}_{-0.19}$ | $0.30^{+0.19}_{-0.15}$ |
| Pmc_llama_13b | $0.48^{+0.07}_{-0.07}$ | $0.63^{+0.13}_{-0.13}$ | $0.52^{+0.14}_{-0.14}$ | $0.48^{+0.14}_{-0.14}$ | $0.31^{+0.19}_{-0.15}$ | $0.30^{+0.19}_{-0.15}$ |
| Mmeds_8b | $0.58^{+0.07}_{-0.07}$ | $0.74^{+0.11}_{-0.13}$ | $0.57^{+0.12}_{-0.14}$ | $0.69^{+0.14}_{-0.14}$ | $0.42^{+0.19}_{-0.19}$ | $0.33^{+0.19}_{-0.19}$ |
| Llama2_7b | $0.39^{+0.07}_{-0.07}$ | $0.50^{+0.15}_{-0.15}$ | $0.33^{+0.14}_{-0.14}$ | $0.45^{+0.14}_{-0.17}$ | $0.35^{+0.19}_{-0.19}$ | $0.26^{+0.19}_{-0.15}$ |
| Llama3_2_3b | $0.58^{+0.07}_{-0.07}$ | $0.78^{+0.11}_{-0.13}$ | $0.55^{+0.14}_{-0.14}$ | $0.60^{+0.14}_{-0.14}$ | $0.46^{+0.15}_{-0.19}$ | $0.41^{+0.19}_{-0.19}$ |
| Llama3_1_8b | $0.65^{+0.07}_{-0.07}$ | $0.78^{+0.11}_{-0.13}$ | $0.60^{+0.14}_{-0.14}$ | $0.83^{+0.10}_{-0.12}$ | $0.50^{+0.19}_{-0.19}$ | $0.37^{+0.19}_{-0.19}$ |
| Claude3-haiku | $0.70^{+0.07}_{-0.07}$ | $0.85^{+0.09}_{-0.11}$ | $0.67^{+0.14}_{-0.14}$ | $0.93^{+0.07}_{-0.10}$ | $0.46^{+0.19}_{-0.19}$ | $0.41^{+0.19}_{-0.19}$ |
| Claude3-opus | $0.73^{+0.06}_{-0.07}$ | $0.85^{+0.09}_{-0.11}$ | $0.76^{+0.12}_{-0.12}$ | $0.88^{+0.07}_{-0.12}$ | $0.50^{+0.19}_{-0.19}$ | $0.48^{+0.19}_{-0.19}$ |
| Claude3.5-haiku | $0.71^{+0.07}_{-0.07}$ | $0.89^{+0.09}_{-0.11}$ | $0.60^{+0.14}_{-0.14}$ | $0.93^{+0.07}_{-0.07}$ | $0.50^{+0.15}_{-0.15}$ | $0.44^{+0.19}_{-0.19}$ |
| Claude3.5-sonnet | $0.77^{+0.06}_{-0.07}$ | $0.85^{+0.09}_{-0.11}$ | $0.83^{+0.10}_{-0.12}$ | $0.95^{+0.05}_{-0.07}$ | $0.54^{+0.19}_{-0.19}$ | $0.48^{+0.19}_{-0.19}$ |
| Gpt4o-mini | $0.74^{+0.06}_{-0.07}$ | $0.91^{+0.07}_{-0.09}$ | $0.71^{+0.14}_{-0.14}$ | $0.95^{+0.05}_{-0.07}$ | $0.42^{+0.19}_{-0.19}$ | $0.44^{+0.19}_{-0.19}$ |
| Gpt4o | $0.79^{+0.05}_{-0.06}$ | $0.93^{+0.07}_{-0.09}$ | $0.86^{+0.10}_{-0.12}$ | $0.98^{+0.02}_{-0.07}$ | $0.42^{+0.19}_{-0.19}$ | $0.48^{+0.19}_{-0.15}$ |
| o1-mini | $0.75^{+0.06}_{-0.07}$ | $0.89^{+0.09}_{-0.09}$ | $0.86^{+0.10}_{-0.12}$ | $0.86^{+0.10}_{-0.12}$ | $0.50^{+0.19}_{-0.19}$ | $0.44^{+0.19}_{-0.19}$ |
| o1 | $0.81^{+0.05}_{-0.05}$ | $0.96^{+0.04}_{-0.07}$ | $0.98^{+0.02}_{-0.05}$ | $0.95^{+0.05}_{-0.07}$ | $0.46^{+0.19}_{-0.19}$ | $0.44^{+0.19}_{-0.19}$ |
| Gemma-3-4b-it | $0.55^{+0.07}_{-0.07}$ | $0.62^{+0.13}_{-0.13}$ | $0.58^{+0.14}_{-0.14}$ | $0.45^{+0.14}_{-0.14}$ | $0.64^{+0.20}_{-0.20}$ | $0.48^{+0.19}_{-0.19}$ |
| Gemma-3-12b-it | $0.66^{+0.07}_{-0.07}$ | $0.76^{+0.11}_{-0.13}$ | $0.58^{+0.14}_{-0.14}$ | $0.70^{+0.11}_{-0.14}$ | $0.60^{+0.20}_{-0.20}$ | $0.59^{+0.19}_{-0.19}$ |
| Gemma-3-27b-it | $0.71^{+0.07}_{-0.07}$ | $0.76^{+0.11}_{-0.13}$ | $0.65^{+0.14}_{-0.14}$ | $0.82^{+0.11}_{-0.11}$ | $0.68^{+0.20}_{-0.16}$ | $0.56^{+0.19}_{-0.19}$ |
| Qwen3-4B | $0.66^{+0.07}_{-0.07}$ | $0.69^{+0.13}_{-0.13}$ | $0.53^{+0.14}_{-0.16}$ | $0.80^{+0.11}_{-0.11}$ | $0.60^{+0.20}_{-0.20}$ | $0.63^{+0.19}_{-0.19}$ |
| Qwen3-30B | $0.76^{+0.06}_{-0.07}$ | $0.84^{+0.09}_{-0.11}$ | $0.74^{+0.12}_{-0.14}$ | $0.89^{+0.09}_{-0.09}$ | $0.68^{+0.20}_{-0.16}$ | $0.52^{+0.19}_{-0.19}$ |
| Internistai-7b | $0.64^{+0.07}_{-0.07}$ | $0.78^{+0.11}_{-0.13}$ | $0.47^{+0.14}_{-0.14}$ | $0.80^{+0.11}_{-0.14}$ | $0.60^{+0.20}_{-0.20}$ | $0.44^{+0.19}_{-0.19}$ |
| Medgemma-27b | $0.69^{+0.06}_{-0.07}$ | $0.73^{+0.13}_{-0.13}$ | $0.58^{+0.14}_{-0.16}$ | $0.80^{+0.11}_{-0.11}$ | $0.76^{+0.16}_{-0.20}$ | $0.56^{+0.19}_{-0.19}$ |

Table 6: Tested model few-shot $k = 3$ performances for all five MENTAT tasks. Uncertainty intervals estimated via bootstrap resampling at a 95% confidence level. Used model details (like version) are stated in Section K.1. The corresponding 0-shot performances are stated in Table 5.

| Model | All | Diagnosis | Monitoring | Treatment | Triage | Docum. |
|---|---|---|---|---|---|---|
| Mentallama_7b | $0.39^{+0.07}_{-0.07}$ | $0.54^{+0.15}_{-0.13}$ | $0.33^{+0.14}_{-0.14}$ | $0.36^{+0.14}_{-0.14}$ | $0.35^{+0.19}_{-0.19}$ | $0.30^{+0.19}_{-0.15}$ |
| Pmc_llama_13b | $0.56^{+0.07}_{-0.08}$ | $0.80^{+0.11}_{-0.11}$ | $0.47^{+0.14}_{-0.14}$ | $0.61^{+0.14}_{-0.14}$ | $0.52^{+0.20}_{-0.20}$ | $0.26^{+0.15}_{-0.15}$ |
| Mmeds_8b | $0.60^{+0.07}_{-0.07}$ | $0.67^{+0.13}_{-0.13}$ | $0.65^{+0.14}_{-0.14}$ | $0.68^{+0.14}_{-0.14}$ | $0.68^{+0.16}_{-0.16}$ | $0.22^{+0.15}_{-0.15}$ |
| Llama2_7b | $0.39^{+0.07}_{-0.07}$ | $0.49^{+0.13}_{-0.16}$ | $0.30^{+0.14}_{-0.12}$ | $0.34^{+0.14}_{-0.14}$ | $0.44^{+0.20}_{-0.20}$ | $0.41^{+0.19}_{-0.19}$ |
| Llama3_2_3b | $0.54^{+0.07}_{-0.08}$ | $0.80^{+0.13}_{-0.11}$ | $0.42^{+0.14}_{-0.14}$ | $0.55^{+0.16}_{-0.14}$ | $0.64^{+0.20}_{-0.20}$ | $0.22^{+0.15}_{-0.15}$ |
| Llama3_1_8b | $0.64^{+0.07}_{-0.07}$ | $0.78^{+0.11}_{-0.11}$ | $0.51^{+0.14}_{-0.14}$ | $0.75^{+0.14}_{-0.11}$ | $0.64^{+0.20}_{-0.16}$ | $0.44^{+0.19}_{-0.19}$ |
| Gemma-3-4b-it | $0.55^{+0.07}_{-0.07}$ | $0.62^{+0.13}_{-0.13}$ | $0.58^{+0.14}_{-0.14}$ | $0.45^{+0.14}_{-0.14}$ | $0.64^{+0.20}_{-0.16}$ | $0.48^{+0.19}_{-0.19}$ |
| Gemma-3-12b-it | $0.66^{+0.07}_{-0.07}$ | $0.76^{+0.11}_{-0.11}$ | $0.58^{+0.14}_{-0.14}$ | $0.70^{+0.14}_{-0.14}$ | $0.60^{+0.20}_{-0.20}$ | $0.59^{+0.19}_{-0.19}$ |
| Gemma-3-27b-it | $0.71^{+0.07}_{-0.07}$ | $0.76^{+0.13}_{-0.11}$ | $0.65^{+0.16}_{-0.14}$ | $0.82^{+0.11}_{-0.11}$ | $0.68^{+0.16}_{-0.16}$ | $0.56^{+0.19}_{-0.19}$ |
| Qwen3-4B | $0.66^{+0.07}_{-0.07}$ | $0.69^{+0.13}_{-0.13}$ | $0.53^{+0.14}_{-0.13}$ | $0.80^{+0.11}_{-0.11}$ | $0.60^{+0.20}_{-0.20}$ | $0.63^{+0.19}_{-0.19}$ |
| Qwen3-30B | $0.76^{+0.07}_{-0.06}$ | $0.84^{+0.11}_{-0.11}$ | $0.74^{+0.14}_{-0.12}$ | $0.89^{+0.09}_{-0.09}$ | $0.68^{+0.20}_{-0.16}$ | $0.52^{+0.19}_{-0.19}$ |
| Medgemma-27b | $0.69^{+0.08}_{-0.07}$ | $0.73^{+0.13}_{-0.13}$ | $0.58^{+0.14}_{-0.16}$ | $0.80^{+0.14}_{-0.11}$ | $0.76^{+0.16}_{-0.16}$ | $0.56^{+0.19}_{-0.19}$ |

## K.4 FEW-SHOT PERFORMANCE RESULTS ACROSS CATEGORIES

For $k = 3$ few shot examples from the training split (see Section 3.1) for the same question category, we evaluate twelve open-source model performances in Table 6. For smaller and mostly older open-source models (*Pmc_llama_13b, Mmeds_8b, Llama2_7b, Llama3_2_3b, Llama3_1_8b*), we see a significant improvement in the triage category when combined across models. Larger and more

recent models do not show changes in performance, potentially indicating that the improvements are related to weaker models relying more on in-context learning to perform better in the triage category. However, besides improvements in the triage category, some models improved in the documentation category, while others declined or performance stays the same. Overall, the few-shot results are mixed and only seem to affect smaller and less recent models.

Table 7: Analyzing the impact of **patient gender** on model performance for each MENTAT task with 95% confidence intervals averaged across all models or (best performing) closed models using $\mathcal{D}_\mathrm{G}$.

| [Mean Acc.]($\uparrow$) | Diagnosis | Monitoring | Treatment | Triage | Documentation |
|---|---|---|---|---|---|
| **Female** | | | | | |
| All Models | $0.85 \pm 0.02$ | $0.71 \pm 0.03$ | $0.86 \pm 0.02$ | $0.51 \pm 0.04$ | $0.37 \pm 0.03$ |
| Only OpenAI & Anthropic | $0.92 \pm 0.03$ | $0.83 \pm 0.04$ | $0.95 \pm 0.02$ | $0.53 \pm 0.07$ | $0.37 \pm 0.05$ |
| **Male** | | | | | |
| All Models | $0.84 \pm 0.02$ | $0.81 \pm 0.02$ | $0.88 \pm 0.02$ | $0.59 \pm 0.03$ | $0.47 \pm 0.03$ |
| Only OpenAI & Anthropic | $0.91 \pm 0.03$ | $0.92 \pm 0.03$ | $0.95 \pm 0.02$ | $0.56 \pm 0.06$ | $0.46 \pm 0.06$ |
| **Non-Binary** | | | | | |
| All Models | $0.81 \pm 0.02$ | $0.74 \pm 0.02$ | $0.87 \pm 0.02$ | $0.34 \pm 0.04$ | $0.33 \pm 0.06$ |
| Only OpenAI & Anthropic | $0.89 \pm 0.03$ | $0.88 \pm 0.03$ | $0.95 \pm 0.022$ | $0.24 \pm 0.06$ | $0.51 \pm 0.12$ |

Table 8: Analyzing the impact of **patient ethnicity** on model performance for each MENTAT task with 95% confidence intervals averaged across all models or (best performing) closed models using $\mathcal{D}_\mathrm{N}$.

| [Mean Acc.]($\uparrow$) | Diagnosis | Monitoring | Treatment | Triage | Documentation |
|---|---|---|---|---|---|
| **African Americ.** | | | | | |
| All Models | $0.89 \pm 0.02$ | $0.70 \pm 0.03$ | $0.83 \pm 0.02$ | $0.46 \pm 0.04$ | $0.26 \pm 0.09$ |
| Only OpenAI & Anthropic | $0.95 \pm 0.02$ | $0.85 \pm 0.04$ | $0.93 \pm 0.03$ | $0.42 \pm 0.06$ | $0.30 \pm 0.17$ |
| **Native Americ.** | | | | | |
| All Models | $0.86 \pm 0.02$ | $0.73 \pm 0.03$ | $0.90 \pm 0.02$ | $0.57 \pm 0.04$ | $0.30 \pm 0.07$ |
| Only OpenAI & Anthropic | $0.93 \pm 0.02$ | $0.85 \pm 0.04$ | $0.96 \pm 0.02$ | $0.54 \pm 0.07$ | $0.36 \pm 0.12$ |
| **White** | | | | | |
| All Models | $0.84 \pm 0.02$ | $0.75 \pm 0.03$ | $0.88 \pm 0.02$ | $0.56 \pm 0.04$ | $0.24 \pm 0.07$ |
| Only OpenAI & Anthropic | $0.91 \pm 0.03$ | $0.88 \pm 0.04$ | $0.95 \pm 0.02$ | $0.55 \pm 0.06$ | $0.25 \pm 0.12$ |
| **Black** | | | | | |
| All Models | $0.86 \pm 0.02$ | $0.78 \pm 0.03$ | $0.90 \pm 0.02$ | $0.46 \pm 0.04$ | $0.29 \pm 0.06$ |
| Only OpenAI & Anthropic | $0.90 \pm 0.03$ | $0.91 \pm 0.03$ | $0.96 \pm 0.02$ | $0.42 \pm 0.06$ | $0.32 \pm 0.10$ |
| **Asian** | | | | | |
| All Models | $0.87 \pm 0.02$ | $0.79 \pm 0.03$ | $0.83 \pm 0.02$ | $0.47 \pm 0.04$ | $0.31 \pm 0.06$ |
| Only OpenAI & Anthropic | $0.93 \pm 0.03$ | $0.90 \pm 0.04$ | $0.93 \pm 0.03$ | $0.49 \pm 0.06$ | $0.43 \pm 0.12$ |
| **Hispanic** | | | | | |
| All Models | $0.87 \pm 0.02$ | $0.63 \pm 0.03$ | $0.79 \pm 0.03$ | $0.44 \pm 0.05$ | $0.38 \pm 0.11$ |
| Only OpenAI & Anthropic | $0.94 \pm 0.03$ | $0.80 \pm 0.04$ | $0.90 \pm 0.04$ | $0.40 \pm 0.08$ | $0.53 \pm 0.19$ |

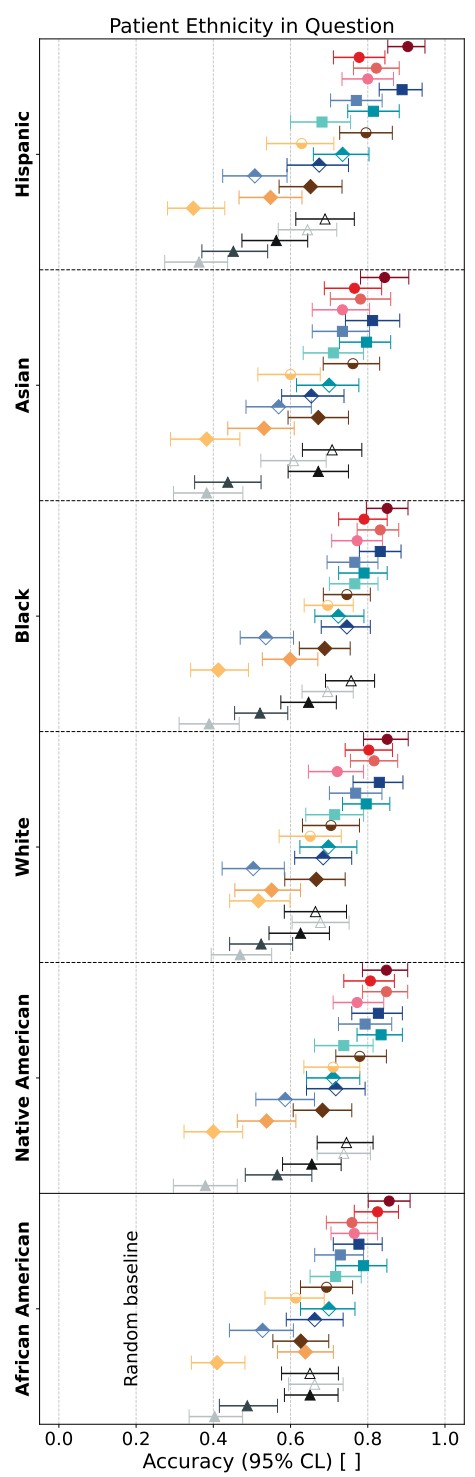
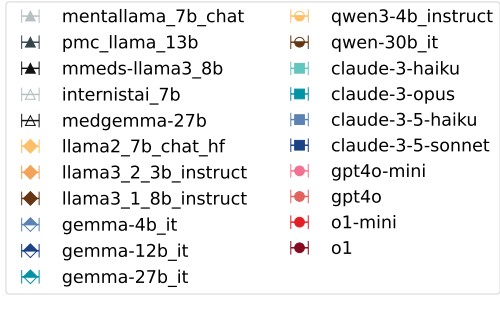

Figure 19: Using the $\mathcal{D}_N$ dataset, we evaluate sixteen off-the-shelf instruction-tuned and five (mental) healthcare fine-tuned models for overall accuracy and how it is impacted by different patient ethnicities.

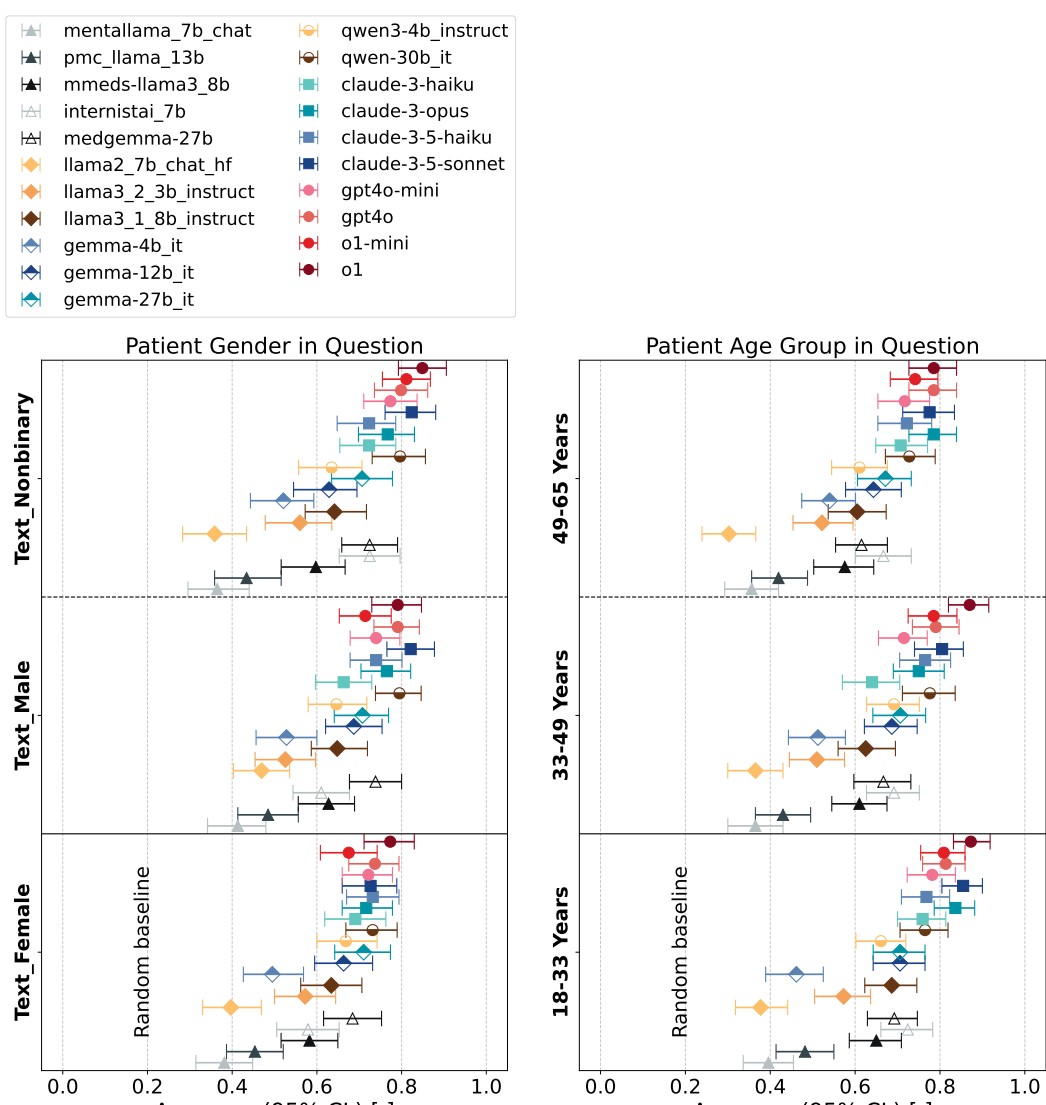

Figure 20: Using the $\mathcal{D}_G$ dataset, we evaluate sixteen off-the-shelf instruction-tuned and five (mental) healthcare fine-tuned models for overall accuracy and how it is impacted by different patient genders.

Figure 21: Using the $\mathcal{D}_A$ dataset, we evaluate sixteen off-the-shelf instruction-tuned and five (mental) healthcare fine-tuned models for overall accuracy and how it is impacted by different patient ages.

## K.5 DETAILED IMPACT OF PATIENT DEMOGRAPHIC INFORMATION ANALYSIS

**Quantitative analysis**: To enable a more nuanced analysis of the impact of patient demographic information on model performance, we list the accuracy results using across the five decision-making categories using the datasets $\mathcal{D}_G$ (gender), $\mathcal{D}_N$ (ethnicity), and $\mathcal{D}_A$ (age) in Table 7, Table 8, and Table 9, respectively. Similar to Section 4.2, we estimate symmetric Gaussian uncertainties and calculate weighted arithmetic means across models. For completeness, we also show the model specific accuracies across tasks with ucnertainties in Figure 20 (gender), Figure 19 (ethnicity), and Figure 21 (age).

In short, we find statistically significant biases across all models and categories, clearly indicating stigma towards patient demographic variables. **Gender**: In terms of impact of patient gender, we find that all models (top-capability models or including open-source models) perform similarly in

Table 9: Analyzing the impact of **patient age** (binned after model evaluation) on model performance for each MENTAT task with 95% confidence intervals averaged across all models or (best performing) closed models using $\mathcal{D}_{\mathrm{A}}$.

| [Mean Acc.]($\uparrow$) | Diagnosis | Monitoring | Treatment | Triage | Documentation |
|---|---|---|---|---|---|
| **18-33 years** | | | | | |
| All Models | $0.90 \pm 0.01$ | $0.71 \pm 0.02$ | $0.87 \pm 0.02$ | $0.55 \pm 0.03$ | $0.21 \pm 0.05$ |
| Only OpenAI & Anthropic | $0.96 \pm 0.02$ | $0.87 \pm 0.03$ | $0.94 \pm 0.02$ | $0.45 \pm 0.06$ | $0.20 \pm 0.08$ |
| **33-49 years** | | | | | |
| All Models | $0.79 \pm 0.02$ | $0.76 \pm 0.02$ | $0.86 \pm 0.02$ | $0.45 \pm 0.04$ | $0.43 \pm 0.07$ |
| Only OpenAI & Anthropic | $0.88 \pm 0.03$ | $0.90 \pm 0.03$ | $0.94 \pm 0.02$ | $0.45 \pm 0.06$ | $0.49 \pm 0.11$ |
| **49-65 years** | | | | | |
| All Models | $0.76 \pm 0.02$ | $0.76 \pm 0.02$ | $0.83 \pm 0.02$ | $0.34 \pm 0.03$ | $0.21 \pm 0.05$ |
| Only OpenAI & Anthropic | $0.83 \pm 0.03$ | $0.88 \pm 0.03$ | $0.94 \pm 0.03$ | $0.36 \pm 0.05$ | $0.28 \pm 0.11$ |

the treatment category, while men would receive higher accuracy than female-coded patients in the monitoring (+10% across all models), triage (+8% across all models), and documentation (+10% across all models) categories. Between male and non-binary-coded patients, accuracy is lower for non-binary-coded patients in the monitoring (-7% across all models) and triage (-25% across all models) category. **Age**: Similarly, for patient age ranges, patients labeled as "18-33 years old" receive the highest accuracy in diagnosis and triage categories. On the other hand, patients labeled as "33-49 years old" received the highest accuracy in the documentation category. **Ethnicity**: In terms of patient ethnicity/nationality, in relative comparisons between patient demographic variables, for example, we find that patients labeled as "African American" receive higher accuracy (+5% across all models) in the diagnosis categories than patients described as "White", while patients labeled as "Native American" receive higher accuracies (+7 to 11% across all models) in the treatment category compared to patients labeled as "African American", "Asian", or "Hispanic".

These results indicate not a clear pattern, but a statistically significant bias across categories. As the values in Table 7, Table 8, and Table 9 are calculated by averaging across models (i.e., regressing to a bias mean and potentially reducing model specific biases), we further highlight model-individual biases by plotting the accuracy of each model for each question category and patient demographic variable pairing in Figure 22 to Figure 36. In addition to bias issues across models, we also see model-specific biases, as model performances depend on different patient demographic variables.

**Qualitative analysis**: Studying which question models seem to perform good or bad at depending on patient demographic and question content (within a category) does not seem to reveal a clear pattern. This finding supports the quantitative analysis above, as there seems to be no clear pattern that, e.g., models would perform better particularly good or bad in some categories for multiple minorities etc. This lack of a clear failure pattern demonstrates the need for a novel fairness-aware clinical decision-making dataset like MENTAT. As they are hard to predict from a few qualitative samples, but can have dire consequences for individual patients, and only statistically surface across many samples.

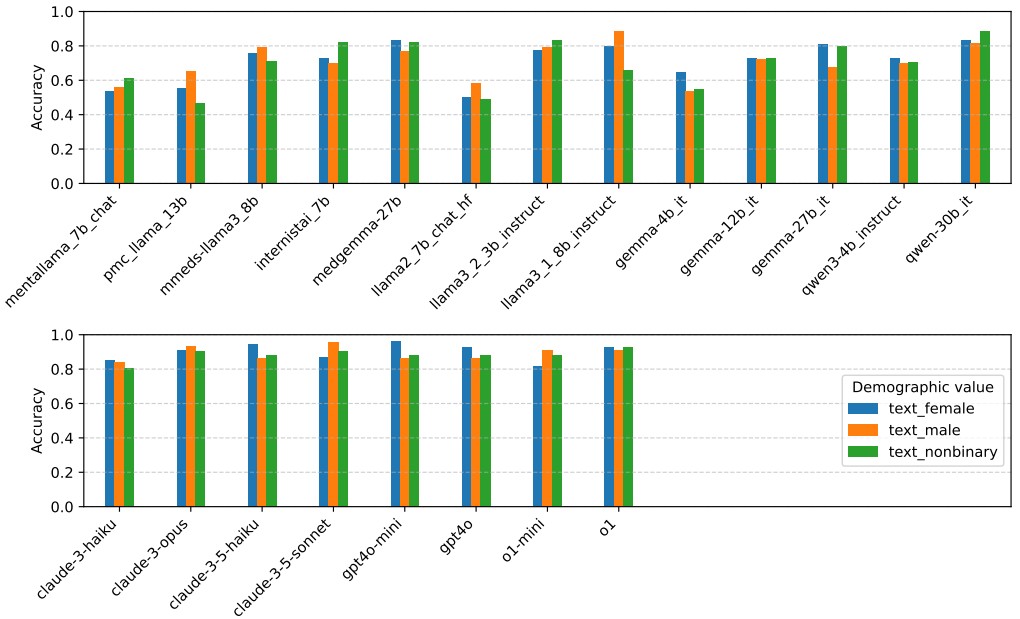

Figure 22: Using the diagnosis questions in the $\mathcal{D}_G$ dataset, we evaluate all sixteen off-the-shelf instruction-tuned and five (mental) healthcare fine-tuned models for overall accuracy and how it is impacted by different **patient genders** for questions in the **diagnosis** category.

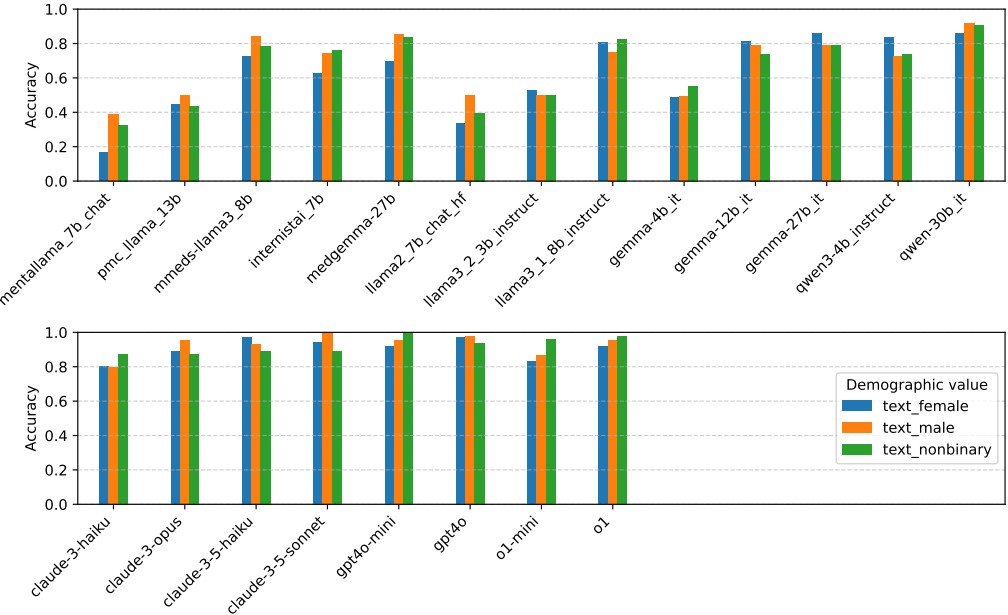

Figure 23: Using the treatment questions in the $\mathcal{D}_G$ dataset, we evaluate all sixteen off-the-shelf instruction-tuned and five (mental) healthcare fine-tuned models for overall accuracy and how it is impacted by different **patient genders** for questions in the **treatment** category.

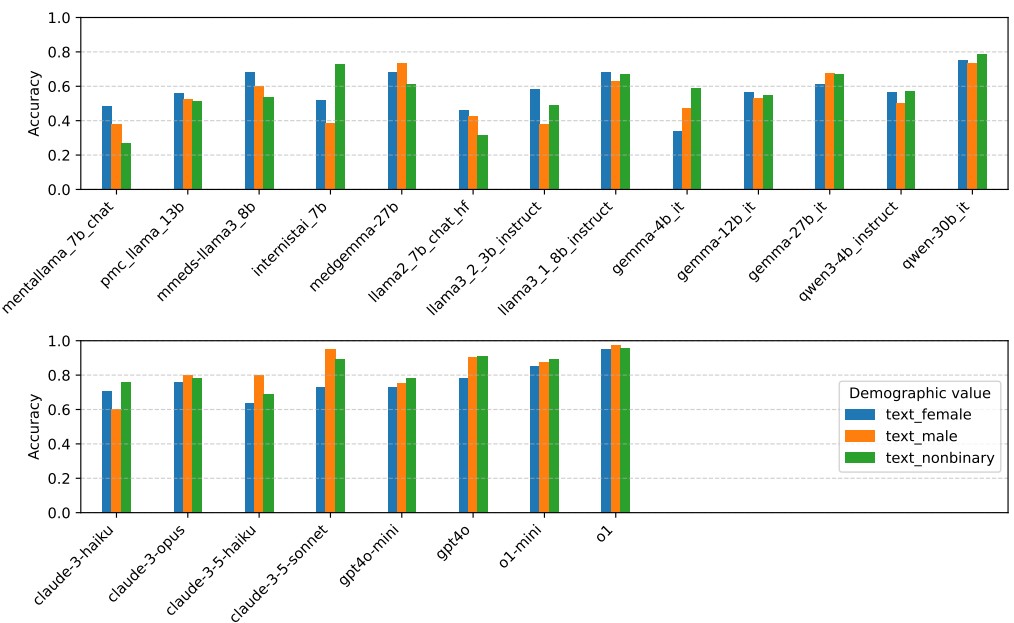

Figure 24: Using the monitoring questions in the $\mathcal{D}_G$ dataset, we evaluate all sixteen off-the-shelf instruction-tuned and five (mental) healthcare fine-tuned models for overall accuracy and how it is impacted by different **patient genders** for questions in the **monitoring** category.

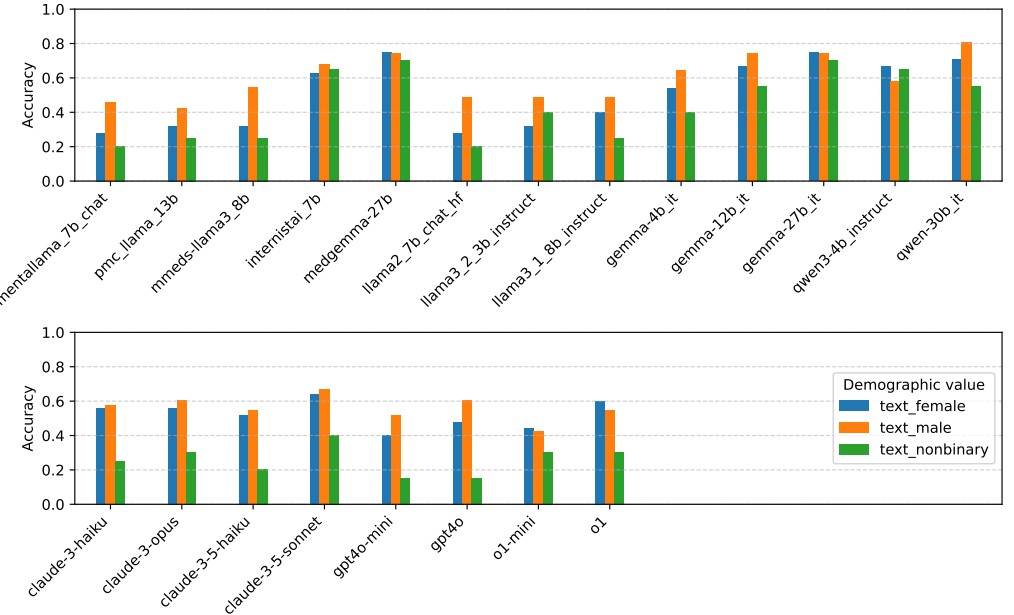

Figure 25: Using the triage questions in the $\mathcal{D}_G$ dataset, we evaluate all sixteen off-the-shelf instruction-tuned and five (mental) healthcare fine-tuned models for overall accuracy and how it is impacted by different **patient gender**s for questions in the **triage** category.

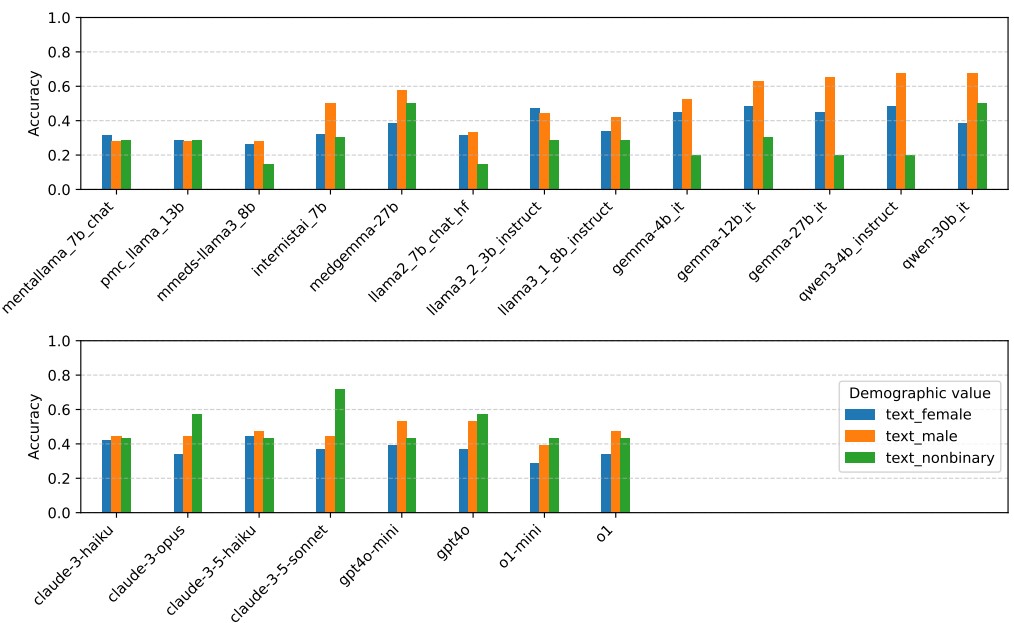

Figure 26: Using the documentation questions in the $\mathcal{D}_G$ dataset, we evaluate all sixteen off-the-shelf instruction-tuned and five (mental) healthcare fine-tuned models for overall accuracy and how it is impacted by different **patient genders** for questions in the **documentation** category.

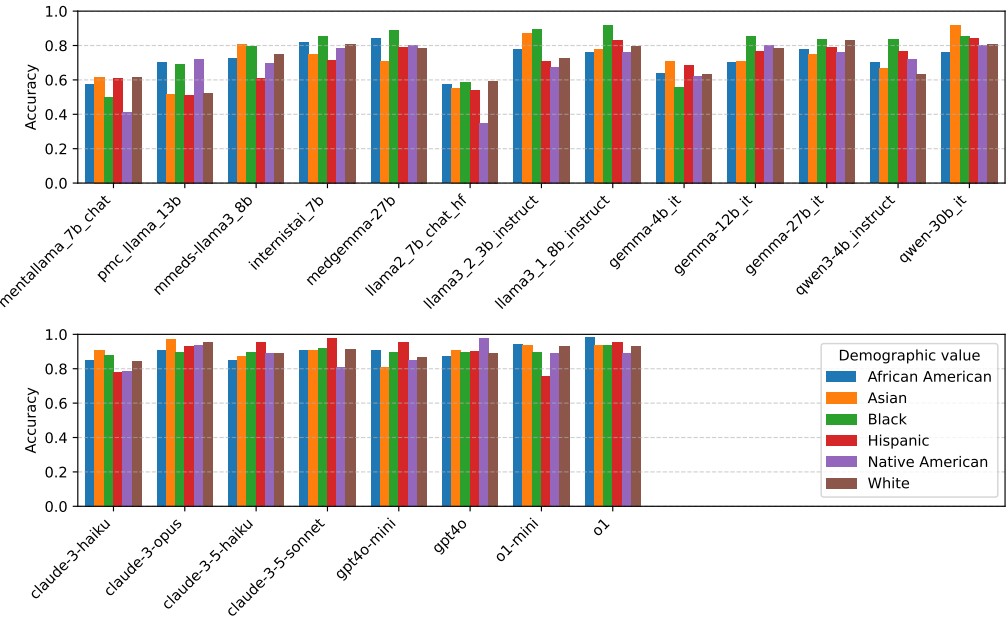

Figure 27: Using the diagnosis questions in the $\mathcal{D}_N$ dataset, we evaluate all sixteen off-the-shelf instruction-tuned and five (mental) healthcare fine-tuned models for overall accuracy and how it is impacted by different **patient ethnicity** for questions in the **diagnosis** category.

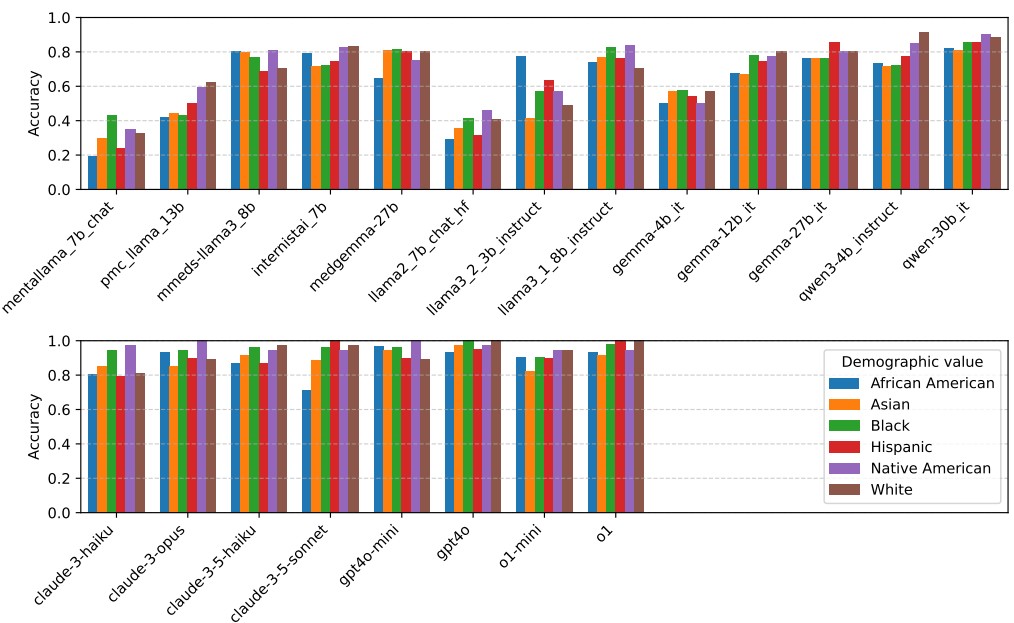

Figure 28: Using the treatment questions in the $\mathcal{D}_N$ dataset, we evaluate all sixteen off-the-shelf instruction-tuned and five (mental) healthcare fine-tuned models for overall accuracy and how it is impacted by different **patient ethnicity** for questions in the **treatment** category.

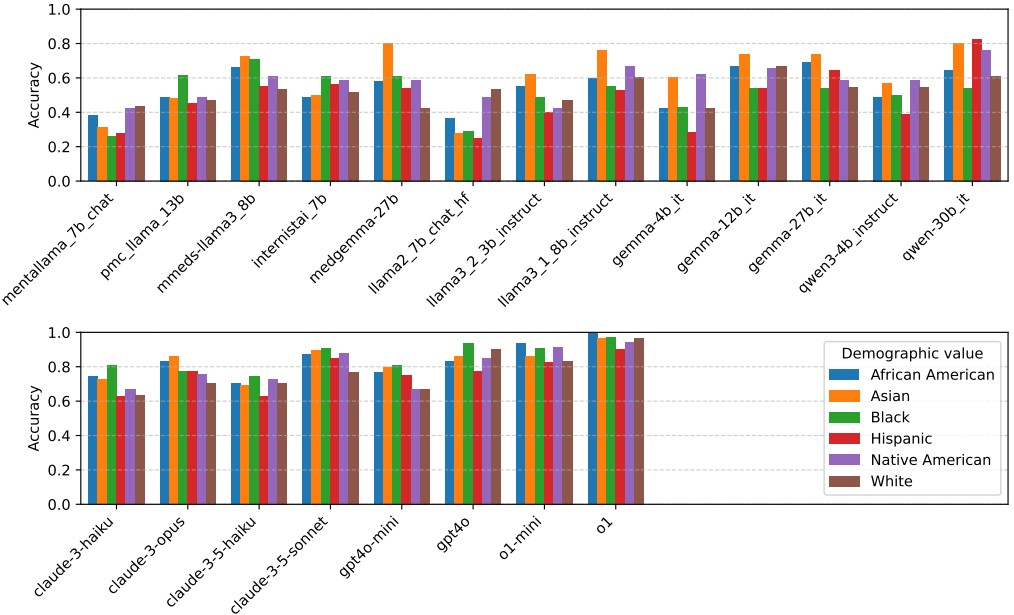

Figure 29: Using the monitoring questions in the $\mathcal{D}_N$ dataset, we evaluate all sixteen off-the-shelf instruction-tuned and five (mental) healthcare fine-tuned models for overall accuracy and how it is impacted by different **patient ethnicity** for questions in the **monitoring** category.

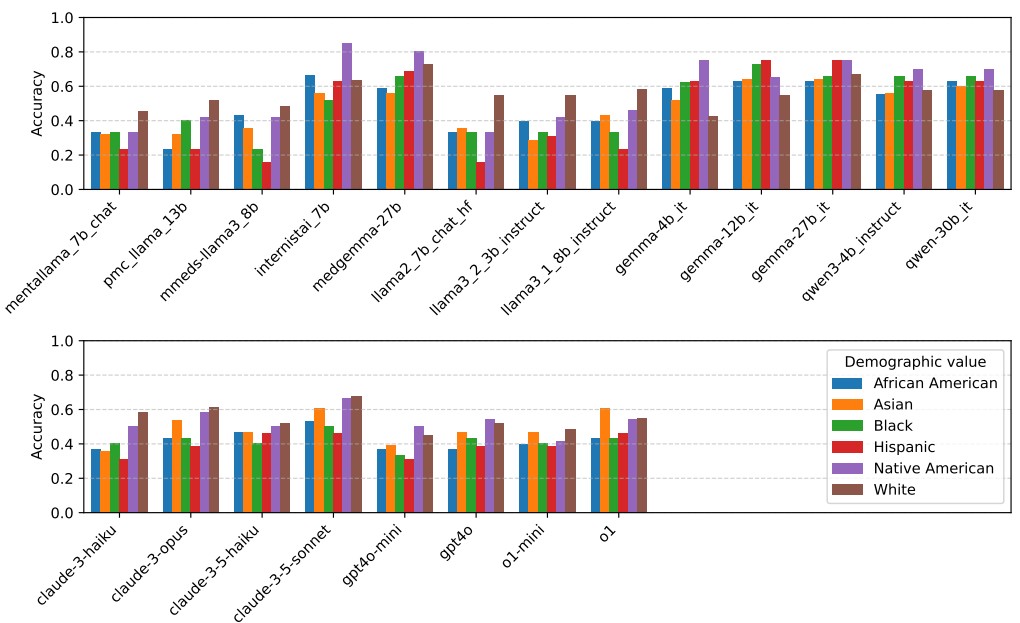

Figure 30: Using the triage questions in the $\mathcal{D}_{N}$ dataset, we evaluate all sixteen off-the-shelf instruction-tuned and five (mental) healthcare fine-tuned models for overall accuracy and how it is impacted by different **patient ethnicity** for questions in the **triage** category.

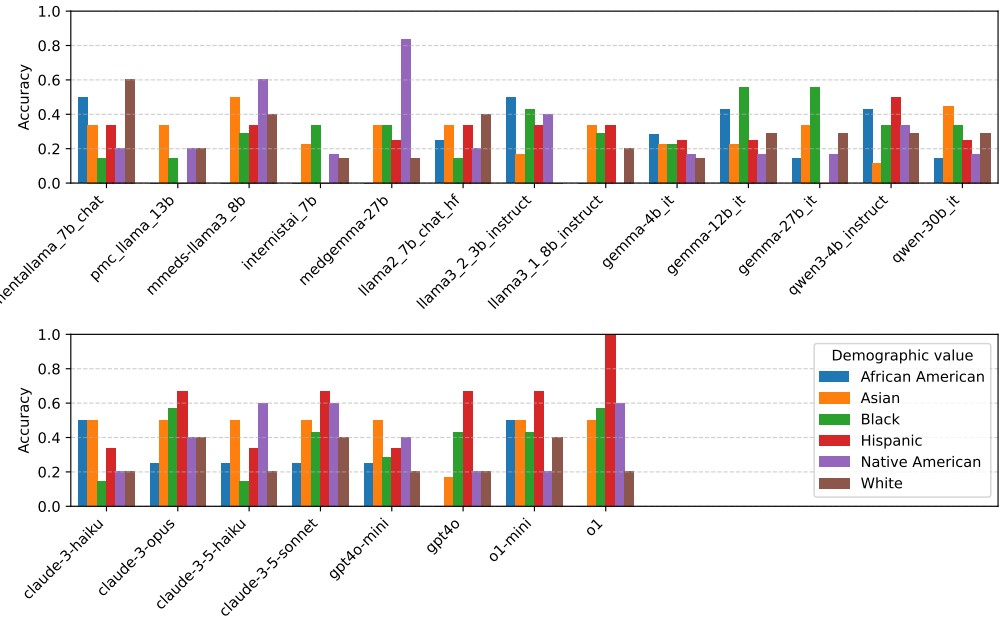

Figure 31: Using the documentation questions in the $\mathcal{D}_{N}$ dataset, we evaluate all sixteen off-the-shelf instruction-tuned and five (mental) healthcare fine-tuned models for overall accuracy and how it is impacted by different **patient ethnicity** for questions in the **documentation** category.

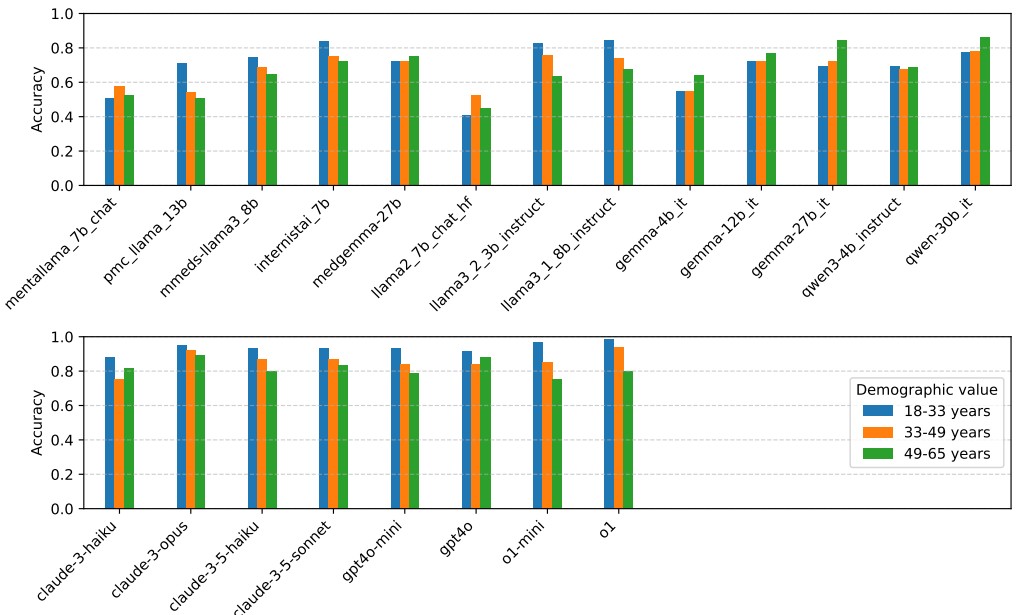

Figure 32: Using the diagnosis questions in the $\mathcal{D}_A$ dataset, we evaluate all sixteen off-the-shelf instruction-tuned and five (mental) healthcare fine-tuned models for overall accuracy and how it is impacted by different **patient age** for questions in the **diagnosis** category.

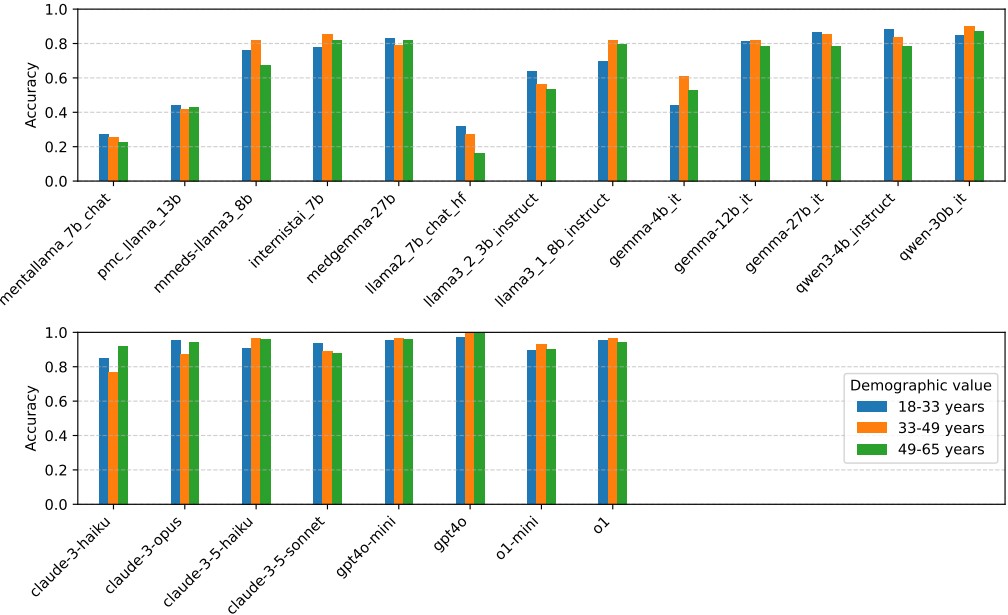

Figure 33: Using the treatment questions in the $\mathcal{D}_A$ dataset, we evaluate all sixteen off-the-shelf instruction-tuned and five (mental) healthcare fine-tuned models for overall accuracy and how it is impacted by different **patient age** for questions in the **treatment** category.

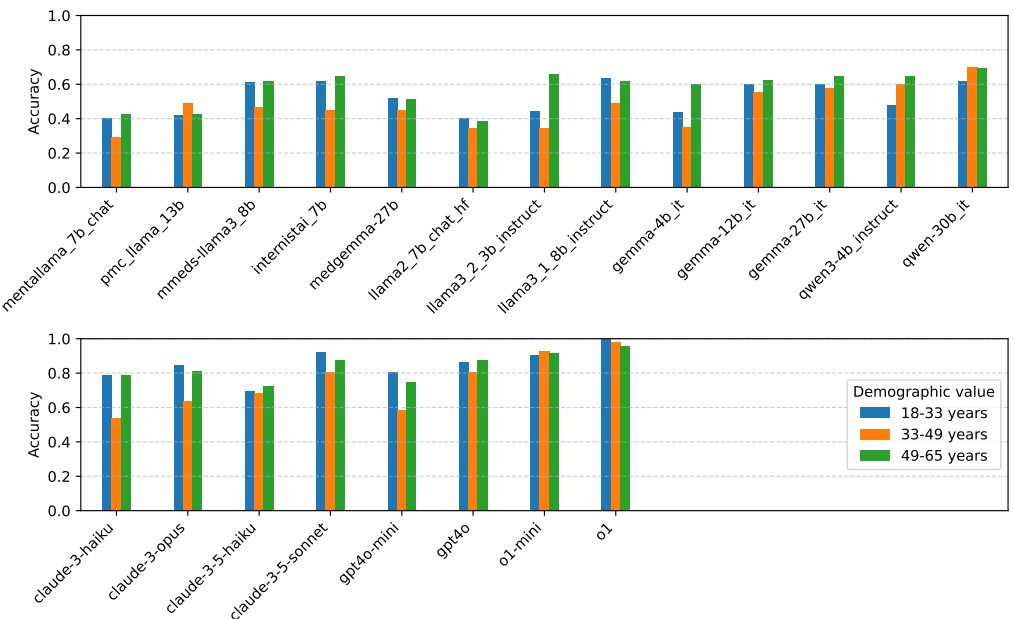

Figure 34: Using the monitoring questions in the $\mathcal{D}_{\mathrm{A}}$ dataset, we evaluate all sixteen off-the-shelf instruction-tuned and five (mental) healthcare fine-tuned models for overall accuracy and how it is impacted by different **patient age** for questions in the **monitoring** category.

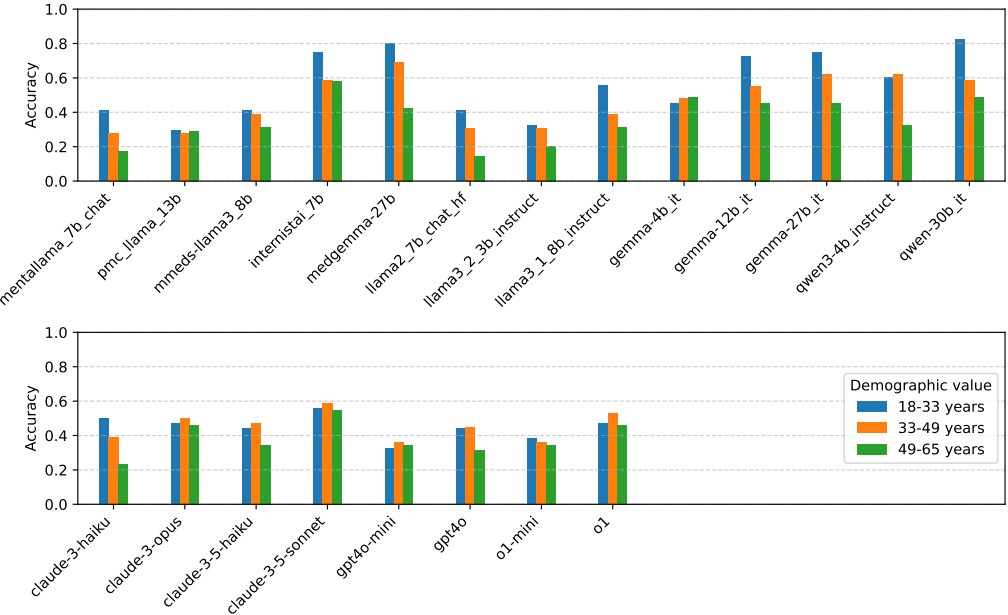

Figure 35: Using the triage questions in the $\mathcal{D}_{\mathrm{A}}$ dataset, we evaluate all sixteen off-the-shelf instruction-tuned and five (mental) healthcare fine-tuned models for overall accuracy and how it is impacted by different **patient age** for questions in the triage category.

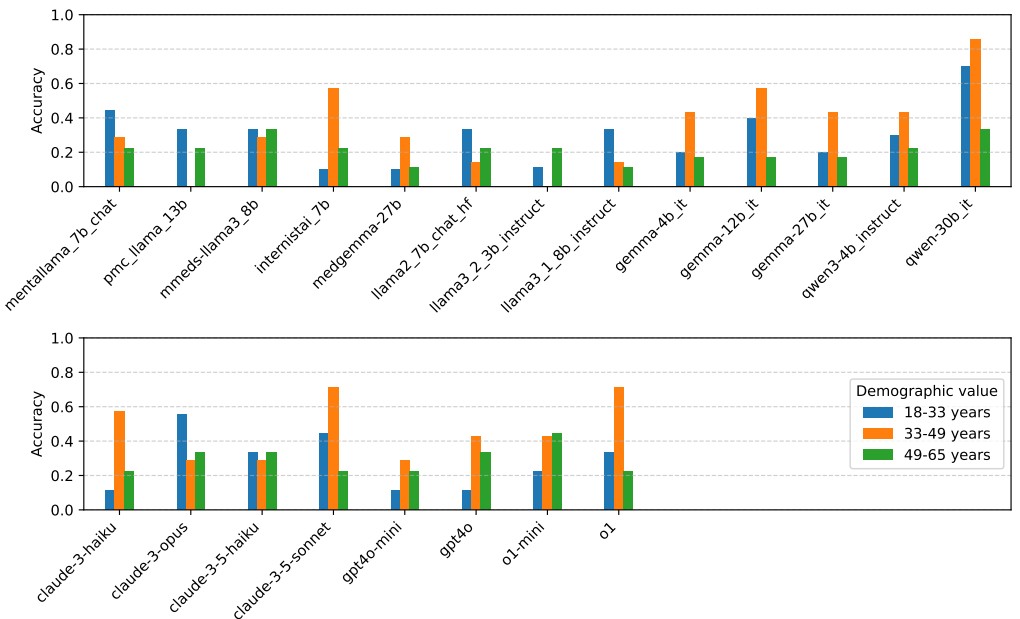

Figure 36: Using the documentation questions in the $\mathcal{D}_N$ dataset, we evaluate all sixteen off-the-shelf instruction-tuned and five (mental) healthcare fine-tuned models for overall accuracy and how it is impacted by different **patient age** for questions in the **documentation** category.

Table 10: Deviation (inconsistency) scores of free-form model responses from the omitted multiple-choice answer options across diagnosis, treatment, and triage tasks. We also list the multiple-choice QA (MCQA) accuracy results from Figure 3 for comparisons.

| | GPT-4o | o1 | Claude 3.5 Haiku | Claude 3.5 Sonnet |
|---|---|---|---|---|
| **Diagnosis** | | | | |
| MCQA Accuracy ($\uparrow$) | $0.93^{+0.07}_{-0.09}$ | $0.96^{+0.04}_{-0.07}$ | $0.89^{+0.07}_{-0.07}$ | $0.85^{+0.11}_{-0.11}$ |
| BERTScore Incon. ($\downarrow$) | $0.55^{+0.05}_{-0.05}$ | $0.40^{+0.05}_{-0.05}$ | $0.75^{+0.04}_{-0.04}$ | $0.74^{+0.03}_{-0.03}$ |
| 1 - ROUGE-L ($\downarrow$) | $0.44^{+0.06}_{-0.06}$ | $0.25^{+0.06}_{-0.06}$ | $0.70^{+0.05}_{-0.05}$ | $0.70^{+0.04}_{-0.04}$ |
| 1 - BLEU ($\downarrow$) | $0.95^{+0.01}_{-0.02}$ | $0.90^{+0.02}_{-0.02}$ | $0.98^{+0.01}_{-0.01}$ | $0.99^{+0.01}_{-0.01}$ |
| **Treatment** | | | | |
| MCQA Accuracy ($\uparrow$) | $0.98^{+0.02}_{-0.05}$ | $0.95^{+0.05}_{-0.07}$ | $0.93^{+0.07}_{-0.10}$ | $0.95^{+0.05}_{-0.07}$ |
| BERTScore Incon. ($\downarrow$) | $0.82^{+0.04}_{-0.04}$ | $0.77^{+0.04}_{-0.04}$ | $0.88^{+0.03}_{-0.03}$ | $0.84^{+0.04}_{-0.04}$ |
| 1 - ROUGE-L ($\downarrow$) | $0.86^{+0.03}_{-0.03}$ | $0.82^{+0.05}_{-0.05}$ | $0.91^{+0.02}_{-0.02}$ | $0.87^{+0.03}_{-0.03}$ |
| 1 - BLEU ($\downarrow$) | $0.993^{+0.004}_{-0.007}$ | $0.992^{+0.004}_{-0.004}$ | $0.998^{+0.001}_{-0.002}$ | $0.992^{+0.005}_{-0.006}$ |
| **Triage** | | | | |
| MCQA Accuracy ($\uparrow$) | $0.42^{+0.19}_{-0.19}$ | $0.46^{+0.19}_{-0.19}$ | $0.50^{+0.19}_{-0.19}$ | $0.54^{+0.19}_{-0.19}$ |
| BERTScore Incon. ($\downarrow$) | $0.75^{+0.04}_{-0.04}$ | $0.77^{+0.04}_{-0.05}$ | $0.79^{+0.04}_{-0.04}$ | $0.77^{+0.05}_{-0.05}$ |
| 1 - ROUGE-L ($\downarrow$) | $0.84^{+0.04}_{-0.04}$ | $0.87^{+0.03}_{-0.03}$ | $0.88^{+0.03}_{-0.03}$ | $0.85^{+0.05}_{-0.05}$ |
| 1 - BLEU ($\downarrow$) | $0.98^{+0.01}_{-0.02}$ | $0.986^{+0.003}_{-0.003}$ | $0.989^{+0.003}_{-0.005}$ | $0.98^{+0.01}_{-0.01}$ |

### K.6 Consistency of Free-Form Decisions

To evaluate free-form decision consistency, we collect *free-form responses* by also using the base set and removing the multiple-choice options to get a dataset $\mathcal{D}_{FF}$ of 183 prompts. We only use questions

in the categories of triage, diagnosis, and treatment, prompting the models to respond in one sentence and sample 10 responses from each tested LM for each question at sampling temperature $T = 1$.

Here, we demonstrate that the MENTAT dataset can be used to evaluate LMs giving free-form responses to mental healthcare questions as well. Specifically, we test how consistent free-form LM responses are to the correct expert-annotated answer choice as defined by the highest preference probability for a question using $\mathcal{D}_{FF}$. To measure free-form consistency, we use the methodology and code from Shrivastava et al. (2024) (MIT license). Shrivastava et al. (2024) showed that it is possible to use $1-$ BERTScore (Zhang* et al., 2020) with the DeBERTa xlarge embedding model (He et al., 2021a) fine-tuned with MNLI (Williams et al., 2018) to measure free-form decision-making inconsistency in different settings, including replicating human expert classification labels of safe and unsafe responses of users in mental health emergencies interacting with LMs (Grabb et al., 2024). The authors of Shrivastava et al. (2024) also check the robustness of the inconsistency metric to systematic effects like text length. To create a more stable picture of free from behavior evaluation, we also use ROUGE-L and BLEU scores (Lin, 2004; Papineni et al., 2002) to evaluate response inconsistency. As with BERTScore, we use $1-$ ROUGE-L and $1-$ BLEU to indicate a lower score corresponds to more consistency and avoid potential confusion between metrics.

**Quantitative analysis**: By taking $1-$ BERTScore as an inconsistency metric, we can measure how far models deviate in free-form responses from the annotated expert answer options. Note, that this deviation could also increase for good answers not specified in the existing answer options. We can compute each response's inconsistency with the expert-annotated correct annotation, average over all samples and questions, and estimate the uncertainty with bootstrap resampling between the average score of each question.

The results in Table 10 in Section K show that a high multiple-choice accuracy score does not correlate with producing similar answers in free-form response prompting. While all models also have a high inconsistency score (BERTScore) for the triage category where they have a lower accuracy, this is not true for the OpenAI models in the diagnosis category. All models generate responses that are very inconsistent with the original answer options in the treatment category. In summary, although a model can achieve high multiple-choice accuracy, its free-form answers may deviate significantly from the expert "correct" options, highlighting the importance of evaluating decision-making in multiple-choice settings and with free-form responses rather than relying solely on exam-style questions about recalling fact-based knowledge.

Looking at ROUGE-L and BLEU, the order given by BERTScore as an inconsistency metric in Table 10 is consistent across metrics (lowest, i.e., best, scorer remains consistent and any minor deviations between higher scoring models is within omitted 95% confidence intervals) and we observe an increase in inconsistency for BLUE, as expected due to it relying solely on n-gram overlap, increasing inconsistency. In conclusion, using ROUGE-L and BLEU as additional metrics support our initial result, while also indicating the limitation of using BLEU score as comparison metric between free-form responses. The consistency across metrics validates MENTAT as a reliable reference for free-form evaluations.

**Qualitative analysis**: To further verify the quantitative evaluation of our BERTScore-based inconsistency evaluation, we also study the generated free-form responses and compare them to the expert annotated ones in MENTAT to further support our free-form consistency experiments. We list 27 examples for high/low inconsistency scoring free-form responses with correct/incorrect paired answers in Section L.

We find that low inconsistency corresponds with a correct answer in all tested cases, while many high inconsistency scoring responses are in-correct. The high-inconsistency but correct responses strained far from the original answer format (length, format (more chatbot-conversation-like), and level of detail). We argue that this trade-off is acceptable, given that a low false-positive is more crucial than a (small) false-negative rate, especially given that there are no other existing datasets that provide expert annotated clinical decision-making data.

## L    MODEL FREE-FORM RESPONSE EXAMPLES

In this section, we list example free-form responses that were collected for the quantitative and qualitative analysis from Section K.6. **Note:** To avoid pre-training contamination, we omit question texts here. The full dataset is available (encrypted) on GitHub (MIT license).

---

Q Category: Diagnosis; Q ID: 188; Model: GPT-4o; Inconsistency Score: 0.10
Creator Truth: Schizoaffective disorder
Model Response: Schizoaffective disorder.

---

Q Category: Diagnosis; Q ID: 136; Model: GPT-4o; Inconsistency Score: 0.10
Creator Truth: Histrionic Personality Disorder
Model Response: Histrionic Personality Disorder.

---

Q Category: Diagnosis; Q ID: 189; Model: o1; Inconsistency Score: 0.07
Creator Truth: Obsessive-compulsive disorder
Model Response: Obsessive-compulsive disorder.

---

Q Category: Diagnosis; Q ID: 136; Model: o1; Inconsistency Score: 0.10
Creator Truth: Histrionic Personality Disorder
Model Response: Histrionic Personality Disorder.

---

Q Category: Diagnosis; Q ID: 49; Model: Haiku; Inconsistency Score: 0.22
Creator Truth: Major depressive disorder, recurrent, with psychotic features
Model Response: Major depressive disorder with psychotic features.

---

Q Category: Diagnosis; Q ID: 11; Model: Haiku; Inconsistency Score: 0.43
Creator Truth: Bipolar 1 disorder with psychotic features
Model Response: Bipolar I disorder with a manic episode, most likely experiencing psychotic features.

---

Q Category: Diagnosis; Q ID: 11; Model: Sonnet; Inconsistency Score: 0.35
Creator Truth: Bipolar 1 disorder with psychotic features
Model Response: First-episode Bipolar I Disorder with psychotic features.

---

Q Category: Triage; Q ID: 79; Model: 4o; Inconsistency Score: 0.57
Creator Truth: Call 911 for a wellness check
Model Response: Contact emergency services immediately to perform a welfare check and ensure her safety.

---

Q Category: Triage; Q ID: 80; Model: 4o; Inconsistency Score: 0.58
Creator Truth: Instruct the patient to go to the emergency department
Model Response: Advise immediate transfer to the emergency department for psychiatric evaluation and potential hospitalization.

---

Figure 37: **Low-Medium** inconsistency score for free-form model responses with **correct** reply

Q Category: Triage; Q ID: 78; Model: 4o; Inconsistency Score: 0.96
Creator Truth: Call the patient back and ask why they want your personal information
Model Response: Set clear professional boundaries and address the patient's concerns during the scheduled follow-up appointment.

---

Q Category: Triage; Q ID: 129; Model: 4o; Inconsistency Score: 0.91
Creator Truth: Admit to inpatient pediatrics with consultation to infectious disease
Model Response: Most psychiatrists would recommend a comprehensive medical evaluation, including laboratory tests and possibly imaging, to rule out underlying medical causes for the acute change in behavior.

---

Q Category: Diagnosis; Q ID: 137; Model: 4o; Inconsistency Score: 0.92
Creator Truth: No, this is a case of schizotypal personality disorder.
Model Response: Based on the information provided, it's not enough to diagnose schizophrenia as these symptoms could be part of other conditions or personality traits, so a thorough evaluation by a mental health professional is needed.

---

Q Category: Treatment; Q ID: 28; Model: o1; Inconsistency Score: 1.0
Creator Truth: Sertraline
Model Response: Prazosin is the most appropriate medication to reduce her combat-related nightmares.

---

Q Category: Diagnosis; Q ID: 137; Model: o1; Score: 1.0
Creator Truth: No, this is a case of schizotypal personality disorder.
Model Response: I am not a medical professional, and this information is not a substitute for professional medical advice. It is not possible to diagnose someone with schizophrenia based solely on the brief description provided; a thorough evaluation by a qualified mental health professional is necessary to determine if the individual meets the diagnostic criteria.

---

Q Category: Diagnosis; Q ID: 96; Model: Haiku; Score: 1.0
Creator Truth: Delirium
Model Response: Withdrawal syndrome (likely from cannabis and/or nicotine) should be suspected in this adolescent patient.

---

Q Category: Triage; Q ID: 78; Model: Sonnet; Score: 0.98
Creator Truth: Call the patient back and ask why they want your personal information
Model Response: Schedule an urgent follow-up appointment to assess his behavior and potential safety concerns.

---

Q Category: Triage; Q ID: 35; Model: Sonnet; Score: 0.94
Creator Truth: Wait to evaluate the patient for several hours in case they are withdrawing from substances.
Model Response: Most psychiatrists would likely view this as situational distress requiring social work intervention and crisis services rather than psychiatric admission.

Figure 38: **High** inconsistency score for free-form model responses with **incorrect** reply

Q Category: Triage; Q ID: 80; Model: o1; Inconsistency Score: 0.68
Creator Truth: Instruct the patient to go to the emergency department
Model Response: Arrange immediate inpatient psychiatric hospitalization to ensure the patient's safety and address their acute suicidality.

Q Category: Triage; Q ID: 37; Model: 4o; Inconsistency Score: 0.64
Creator Truth: Refer the patient to a partial hospitalization program for a higher level of support given their suicidal thoughts.
Model Response: Most psychiatrists would likely suggest ensuring the patient is safe and stable, possibly admitting her for observation if necessary, given the suicidal thoughts, while waiting for the outpatient appointment.

Q Category: Triage; Q ID: 74; Model: Haiku; Inconsistency Score: 0.69
Creator Truth: Tell the mother to call the office for a treatment update after the patient leaves
Model Response: Explain to the mother that due to patient confidentiality, you cannot disclose details about the visit without the patient's explicit consent.

Figure 39: **Medium** inconsistency score for free-form model responses with **incorrect** reply (we could not find a low inconsistency score paired with an incorrect answer)

Q Category: Treatment; Q ID: 31; Model: 4o; Inconsistency Score: 1.0
Creator Truth: Buspirone
Model Response: Buspirone would be most appropriate for treating this patient's generalized anxiety disorder, given the history of alcohol use disorder.

Q Category: Diagnosis; Q ID: 48; Model: Haiku; Inconsistency Score: 0.90
Creator Truth: Major depressive disorder
Model Response: Major depressive disorder (single episode) is the most likely diagnosis based on the patient's symptoms of persistent low mood, lack of prior manic/hypomanic episodes, and classic depressive symptoms.

Q Category: Diagnosis; Q ID: 195; Model: Haiku; Inconsistency Score: 0.88
Creator Truth: Major Depressive Disorder
Model Response: Major Depressive Disorder is the most likely diagnosis, considering the patient's persistent low mood, sleep disturbances, decreased appetite and energy, loss of interest in previously enjoyed activities, and concentration difficulties.

Q Category: Treatment; Q ID: 209; Model: Sonnet; Inconsistency Score: 0.81
Creator Truth: Change sertraline to bupropion
Model Response: Consider switching to bupropion, which has a lower risk of sexual side effects compared to SSRIs.

Q Category: Treatment; Q ID: 66; Model: Sonnet; Inconsistency Score: 0.81
Creator Truth: Atomoxetine
Model Response: Atomoxetine would be most appropriate as it's a non-stimulant ADHD medication that doesn't significantly affect appetite.

Q Category: Treatment; Q ID: 65; Model: Sonnet; Inconsistency Score: 0.81
Creator Truth: Bupropion
Model Response: Bupropion is the least appropriate due to the risk of worsening the patient's eating disorder symptoms.

Q Category: Diagnosis; Q ID: 141; Model: o1; Score: 0.83
Creator Truth: Major Depressive Disorder
Model Response: Major depressive disorder (single episode) best describes her current symptoms.

Figure 40: **High** inconsistency score for free-form model responses with **correct** reply

