# OpenReview forum: "Moving Beyond Medical Exams: A Clinician-Annotated Fairness Dataset of Real-World Tasks and Ambiguity in Mental Healthcare"
_ICLR.cc/2026/Conference — ICLR 2026 Poster_

### Official Review · Reviewer_snwZ · 2025-10-24

**Soundness:** 3
**Presentation:** 2
**Contribution:** 2
**Rating:** 6
**Confidence:** 5

**Summary:**

This paper presents a novel evaluation MENTAT designed to assess the performance on LMs for psychiatry. The authors introduce five dimensions "diagnosis", "treatment", "triage", "monitoring", and "documentation" to cover a more realistic range of challenges faced by psychiatrists in real-world clinical practice. The dataset is annotated by experts with a sound disagreement resolution method. This led to a dataset mainly used a multiple choice question dataset. The paper then evaluates the performance of a variety of models and unsurprisingly demonstrate the state-of-the-art models achieve impressive the best results. The five dimensions show discrepancy in terms of capabilities with every model achieving low scores on documentation performance while SOTA models achieve near perfect scores on treatment.

**Strengths:**

The paper fills a gap in current evaluations by addressing psychiatry and especially given the 5 dimensions. The involvements of clinical experts strengthens the relevance and reliability of the evaluation compared to common approaches which are sourced from old exams or synthetic data generated by LLMs.

As said in the summary, the disagreement resolution method is solid and well motivated with a sufficient number of annotations.

The results are clearly communicated with appropriate statistical analysis.

Introducing variations in age, gender, and ethnicity is a welcome addition with a clear discussion of the impact and reinforces previously identified biases in LLMs.

**Weaknesses:**

The framing of the evaluation in regards to related work is ambivalent. On the one hand the authors make the case that current board style methods are insufficient but they then proceed to build a board style exam using domain specific cases. The limitations of MCQs especially for LLMs are not addressed appropriately in the introduction/related work.

The experiment tests small models and SOTA models but do not include mid-sized models or specialized medical models that have demonstrated measurable performance increase over the base models, for example PMC Llama was not instruct tuned where as other models like GPT-4o are instruction tuned. In addition, the absence of a human baseline makes interpretation of the results difficult. For USMLE based benchmarks for example we know that the passing score is around 60% and that students score around 70%.

Presentation could be improved, Figure 3 especially is difficult to read due to the usage of colors that are too similar.

As the authors explicitly address, this benchmark is limited by being US centric and having a relatively small number of samples. However, the authors appropriately justify that evaluations specific to local practices is important for clinical relevance and that curating evaluations with experts is a difficult and expensive process. While I agree with both arguments, my concern is with the choice of venue, it appears to me that the dataset's primary goal is to be clinically informative for US clinical settings. I wonder if a medical journal such as JAMA Psychiatry would be more appropriate and find the correct audience.

**Questions:**

# Major concers
1) There seems to be a contradiction in the introduction, board-style exams are said to be limited because they have only one correct answer but then the MENTAT dataset is transformed into board-style questions with a MCQ format with 5 options with only one correct. Can the author clarify their position on the place of MCQs? Maybe also include a discussion of different techniques such as LLM-as-a-judge and key features.

2) The related work section could be expanded to include previous work on biases found in models [1] as well as the limitations of the multiple choice method for evaluating LMs which the paper also describes when comparing the performance on open-ended questions [2].

3) There is a major gap in size and performance between the open models and the closed models evaluated. Most open models included are either outdated (PMC Llama) or small. I would have liked to see high performing open models like Qwen3, Gemma3 in different sizes (from small 3b to larger 200b+ models) and specialized models like MedGemma3, Internist.ai, Meditron3.

4) While the dataset is annotated by experts, a baseline score for 1 or 2 psych residents/attendings would allow readers to better understand what a score of 50% on documentation represents. Are the models failing on this task compared to humans? Or would humans get similar results?

# Minor suggestions

1) Line 168: "are practitioners and M.D.s", could you detail the distribution? Including seniority level (residents vs attendings, MD/APP/NP)
2) The 95% CI is included in every graph but I could not find the exact calculation used to compute them. Could you clarify?
3) Figure 3 and Tables 1 and 2 arrive a bit too early in the document giving it a less polish feel and impeding comprehension, I suggest moving them down a page or two.
4) In Figure 3, the color code and shapes are difficult to differentiate in the figure, for instance I am unable to tell if pmc_llama is better than mmeds in the figure because I can't tell one from the other. This issue is present for multiple pairs (o1/o1-mini for example).

# References:
[1] Assessing the potential of GPT-4 to perpetuate racial and gender biases in health care: a model evaluation study (Zack et al. The Lancet Digital Health 2024)

[2] Pattern Recognition or Medical Knowledge? The Problem with Multiple-Choice Questions in Medicine (Griot et al., ACL 2025)

---

> ### Author Response · Authors · 2025-11-26
>
> Dear Reviewer 4 (snwZ),
>
> Thank you for your thorough review and constructive feedback! We appreciate your recognition of our expert annotation methodology and statistical rigor. We address your concerns below and have conducted substantial new experiments.
>
> ## Regarding Major Concerns (MC)
>
> **MC1**:  MCQ Format vs. Board Exams
>
> We appreciate the opportunity to clarify this important distinction. Our criticism of board exams centers on their emphasis on fact recall rather than their multiple-choice format. MENTAT's MCQ format serves a fundamentally different purpose. Board exams test: "What is the mechanism of action of lithium?" (memorization), while MENTAT tests: "Given this patient's symptoms, treatment history, and side effects, what medication adjustment is most appropriate?" (clinical decision-making).
>
> The MCQ format is a necessary first step for several reasons. First, benchmark calibration, as alternative methods (LLM-as-judge, automated classifiers) must first be validated against human expert ground truth, which only MENTAT currently provides for clinical psychiatry. Second, the alternatives have know limitations. For example, LLM judges exhibit position bias, verbosity bias, and require calibration datasets (which don't exist for psychiatric decision-making). Also, our results show that synthetic dataset approaches currently do not solve this issue, further highlighting the need for MENTAT. Finally, MCQs enable controlled fairness evaluation (Table 2) and reliable statistical analysis.
>
> Our free-form consistency analysis (Section F.1, expanded with ROUGE-L and BLEU see general comment "Exp 1") demonstrates MENTAT's utility beyond MCQs. We will strengthen our discussion of MCQ limitations vs. benefits in the revised manuscript and expand related work to include [Griot et al., ACL 2025] and bias literature [Zack et al., 2024].
>
> **MC2**: Model Coverage: Mid-Sized & Specialized Models
>
> We have now evaluated 7 additional models (see general comment "Exp 3"), based on your suggestions. We find that recent open models approach closed-model performance on triage/documentation but lag on diagnosis/treatment. Also, specialized medical models show improvement in specific areas but don't uniformly outperform strong general models. Combined, these results validate MENTAT's discriminative power and highlights that generic medical pre-training doe snot guarantee psychiatric decision-making competence
>
> **MC3**: Human Baseline
>
> MENTAT is the human expert baseline (100% by construction, majority vote). Unlike USMLE (designed for students), our questions were created and verified by practicing psychiatrists specifically to reflect expert-level decision-making. The clinically relevant comparison is whether models match expert consensus. Our results (Table 1 and the "Exp 3" in the general rebuttal comment) show even SOTA models do not achieve 100% revealing critical safety gaps.
>
> **MC4**: ICLR vs. Medical Journal
>
> We respectfully, but strongly, disagree and argue ICLR is the ideal venue for three reasons: Technical ML contributions, broader AI research value, and target audience.
>
> Technical ML contributions: Our used methodology with the hierarchical Bradley-Terry preference modeling with annotator-specific parameters (Eq. 2-3), rigorous uncertainty quantification through bootstrap resampling, and free-form consistency evaluation methodology (Section F.1) are too technical for medical journals.
>
> Broader AI research value: MENTAT provides a domain-specific testbed for ML problems beyond psychiatry, as it covers preference learning with inherent ambiguity (not just generic toxicity/helpfulness), fairness evaluation under demographic variation (templates enable controlled studies), and human expert disagreement modeling.
>
> Target audience: ICLR attendees developing alignment methods, fairness interventions, and medical AI systems need psychiatric evaluation tools. Medical journals lack the ML expertise to assess our technical contributions (HBT modeling, statistical methodology) and wouldn't reach researchers building LLMs.
>
> Finally, MENTAT successfully covers multiple relevant topics of the ICLR call for papers.
>
> ## Regarding Minor Concerns
>
> * 95% CI calculation: Bootstrap resampling (1000 iterations, stated in Section 4.1. We will better highlight the details)
> * Figure 3 presentation: We chose the color palette to be suitable for colorblind readers, but we will
> (1) add a table with all details (models, categories, uncertainties, ...) similar to the one in "Exp 3" in the general rebuttal comment, (2) enlarge Figure 1, (3) test alternative colorblind-safe palettes while preserving symbol redundancy
> * Table/Figure placement: Yes, we will adjust for improved flow.
>
> We believe these additions substantially strengthen the paper and address all your concerns. Would you be willing to reconsider your scores given these improvements?
>
> Thank you for considering MENTAT for ICLR.
> Best, the authors

---

> > ### Comment · Reviewer_snwZ · 2025-11-26
> >
> > Dear authors,
> >
> > Thank you for your detailed response.
> >
> > I appreciate your work and commend you for the additional experiments, which I believe strengthen your already strong submission.
> >
> > I have raised my score accordingly.
> >
> > As a side note, regarding the venue choice, I agree with your arguments. Beyond this ICLR submission, I recommend that you also submit a more clinically oriented version of your work to share your findings with a broader audience, as few MDs attend ML conferences.

---

> > > ### Author Response · Authors · 2025-11-26
> > >
> > > Thank you!
> > >
> > > On MDs attending ML conferences, that is a great point and you are right. We will look for venues beyond the ML community.
> > >
> > > Best, the authors

---

### Official Review · Reviewer_3LdC · 2025-10-26

**Soundness:** 3
**Presentation:** 2
**Contribution:** 3
**Rating:** 4
**Confidence:** 3

**Summary:**

This paper pioneers a new evaluation dataset for LLMs in mental health care: MENTAT. The authors curate a fully human written and annotated set of questions with multiple choice answers across five categories in mental health. Novelly, answer choices are not strictly correct and incorrect. Annotator preference is used as a proxy for simulating multiple correct answers and evaluated against. The authors present results of state-of-the-art LLMs on their dataset demonstrating specific, actionable performance gaps.

**Strengths:**

-	A fully human generated dataset is an exciting contribution.
-	The dataset’s multiple category framework provides an informative lens into which aspects of mental health are falling behind in current LLMs, which is very informative and actionable. Models perform less well in triage and documentation domains, which is an important insight.
-	Comparison of questions across demographic variables is clever, informative, and comprehensive.
-	The inclusion of preference annotations and multiple correct answers is nice technique that is likely applicable in LLM evaluation in medicine more broadly.

**Weaknesses:**

-	The description of the related work in the mental health space is confusing. It would be helpful to contrast which of these studies introduce benchmarks and to state an overall understanding of the state of LLM performance on mental health tasks. What are the generally accepted specific gaps or are they unknown?
-	Appendix plots of model performance across demographic variables do not suggest significance for the differences across them within groups. This is confusing given the claims in the text of better performance in some groups. It is also confusing how narrow the confidence intervals are in Table 1, 2 compared to in the Figures (ie 3, 19).
-	The overlap of methods and results in text in section 3.2 makes the section dense and challenging to read. It remains unclear whether evaluation occurs against modeled probabilities or raw annotation preferences.
-	It would be worth further explanation in the text of why comparisons with the Shrivastava et al BERTScore metric for comparing the free form text is valid as this is a core suggested contribution.

**Questions:**

-	On language, how is MENTAT not a benchmark? I see how the dataset is a novel benchmark but not the contrast between “evaluation dataset” and benchmark.
-	How are questions written by experts? Do experts create cases from real patient cases, study texts, or are scenarios entirely contrived?
-	How did you get 11.5 annotations per question with 8 annotators?
-	Which accuracy comparisons are computed using one of the Bradley Terry models versus the raw preference probabilities or annotation scores?

---

> ### Author Response · Authors · 2025-11-26
>
> Dear Reviewer 3 (3LdC),
>
> Thank you for your thoughtful review and recognition of MENTAT's contributions! We appreciate your positive assessment of our human-generated dataset, multi-category framework, and demographic analysis. Below, we address our concersn:
>
> **Weakness 1**: Related work clarity in mental health LLM benchmarks
>
> Indeed, you raise an important point. There are no existing clinician-annotated, real-world decision-making benchmarks for mental health. Still, we will better highlight and differentiate this in our related work section. Existing work covers exam-style factual recall (MedQA, MMLU), therapy chatbots, or automated documentation, but MENTAT evaluates clinical decision-making with expert-annotated ambiguity and fairness analysis. We will restructure Section 2 to explicitly contrast these and look for additional sources, if applicable.
>
> **Weakness 2** Statistical significance of demographic differences
>
> We agree that the plots do not capture the full differences and nuances from the demographic fairness analysis. We will add tables with the nuemrical details to the appendix, see also our results in "Exp 2" in the general comment. As with the initial submission, we find that different minorities are disadvantaged in different categories. This reveals "random discriminatory bias", only surfaced by statistical analysis but unpredictable for individual (qualitative) data points, highlighting the importance of MENTAT.
>
> Also, the confidence intervals differ as Tables 1-2 aggregate across questions and models with estimated Gaussian uncertainties, narrowing the confidence intervals.
>
> **Weakness 3**: Section 3.2 clarity
>
> We see where you are coming from and w'll split Section 3 into two sections: Dataset Design (creation, categories) and Annotation Methodology (preference model, Bradley-Terry). To calrify, all evaluations use Bradley-Terry modeled probabilities (Equation 2), not raw annotations, to normalize annotator-specific biases, which is validated by Figure 2.
>
> **Weakness 4**: BERTScore validation for free-form evaluation
>
> Good point! There are three lines of validating our free-form evaluation and using BERTScore.
> * Shrivastava et al. (2024) already validated BERTScore for mental health expert annotation replication and tested for systematic biases (e.g., text length correlation). However, to further validate our approach, we replicated their results to check our implementation using the synthetic data in Shrivastava et al. (2024) and the mental health expert annotated LM responses from Grabb et al. (2024) with our implementation.
> * We conducted new multi-metric validations, see "Exp 1" in the general rebuttal comment. Besides BERTScore, we also use ROUGE-L and BLEU to confirm model consistency ordering, while also validating BERTScore as better choice due to BLEU's higher inconsistency from reliyng on n-grams.
> * Finally, we also conduct a qualitative analysis to support our approach, see "Exp 1" in the general rebuttal comment.
>
> **Question 1**: Why "evaluation dataset" not "benchmark"?
>
> To be fair, you are right. This is probably more confusing than necessary. We believe that a lot of benchmarks lack the necessary construct validity to serve as meaningful benchmark and would probably be better labeled as eval datasets. We will adopt "benchmark" terminology.
>
> **Question 2**: How are questions written?
>
> Questions are written from clinical experience (not real cases/textbooks). Experts draft realistic scenarios, at least one second expert verifies proposed scenario, and any conflicts are resolved via discussion (all 9 annotators are practicing M.D. psychiatrists).
>
> **Question 3**: 11.5 annotations per question with 8 annotators?
>
> The questions were randomized during the annotation process, so some questions got more annotations. This is statistically valid, as it only leads to narrower confidence intervals and more signal for individual questions for the Bradley-Terry model with no systematic bias. As the annotations were conducted over time, some annotators annotated the same question multipel times (but with slightly different scores, capturing crucial nuances where a clinical decision can also depend on the day (we thought about this a lot)).
>
> **Question 4**: Bradley-Terry vs raw scores?
>
> All accuracy evaluations use the Bradley-Terry probabilities and raw scores are only shown for transparency in Figure 1. BT used only for triage/documentation, as other categories have single correct answers (creator-verified), so no preference modeling needed.
>
> **Summary:** We have addressed all concerns with new experiments validating free-form evaluation, revealing nuanced demographic biases, proposed structural improvements, and methodological clarifications. We believe these revisions significantly strengthen the paper's clarity and contribution. Thank you for your feedback!
>
> Best, the authors

---

### Official Review · Reviewer_W1d3 · 2025-11-08

**Soundness:** 3
**Presentation:** 3
**Contribution:** 3
**Rating:** 6
**Confidence:** 4

**Summary:**

The paper introduces MENTAT, a clinician-authored evaluation dataset for psychiatric decision-making that moves beyond exam-style QA. It covers five domains (diagnosis, treatment, monitoring, triage, and documentation) with 203 base vignettes expanded via demographic variables (age, gender coding, ethnicity) to probe fairness. Ambiguous categories (triage, documentation) receive expert preference labels derived with a hierarchical Bradley-Terry (HBT) model. The authors evaluate 11 general LMs and several health-tuned LMs; models score high on diagnosis/treatment, but struggle on triage/documentation, and show sensitivity to demographic attributes.

**Strengths:**

1. Clinician-first design with verification and a thoughtful ambiguity pipeline that avoids LM-generated contamination during creation.

2. Methodological clarity: transparent HBT formulation (annotator effects), bootstrap CIs, and explicit prompt templates.

3. Writing is clear and is generally easy to follow.

**Weaknesses:**

1. Post-hoc zeroing of 'objectively wrong' answers (then renormalizing) encodes expert priors that bypass the HBT inference. The paper states most zeroed options had p ≤ 0.2 anyway, but no sensitivity analysis quantifies how many questions' top-ranked answer changes under alternative thresholds or no clamping. This risks conflating empirical disagreement with ex-post correctness.


2. While Section C justifies jurisdiction-specificity, the abstract/intro should state this limitation upfront.

**Questions:**

1. 203 items may suffice for ranking models but are too few for fine-tuning or error analysis by symptom subtype. Did the authors consider releasing raw clinician rationales (sanitized) to enable richer qualitative studies??

2. Could you include ablations for answer-order randomization and few-shot vs zero-shot?

3. Beyond average accuracy, did you analyze error types by demographic condition? That would better ground fairness risks.

4. The paper states "diverse clinicians" but doesn't report demographics of the 9 annotators themselves. Could annotator bias (e.g., age, geography, years in practice) confound the HBT parameters?

---

> ### Author Response · Authors · 2025-11-26
>
> Dear Reviewer 2 (W1d3),
>
> Thank you for your constructive review and recognition of MENTAT's clinician-first design and methodological clarity. Below, we address our concersn:
>
> **Question 3**:  Error types by demographic condition for fairness analysis
>
> Great point! We conducted exactly this analysis in response to your review, see "Exp 2" in the general rebuttal comment. We decomposed accuracy by demographic variable and category to identify where biases concentrate. In a quantitative analysis, we find statistically significant differences for category-specifc performance depending on patient demographic information, but without clear pattern. We also verify the lack of a predictable pattern with a qualitative analysis. Our results indicate that these are not stereotypical biases (e.g., "model always fails for demographic  group X") but random discriminatory biases that surface only statistically across many samples. This makes them more harder to detect without datasets like MENTAT.
>
> **Weakness 1**: Post-hoc zeroing of objectively wrong answers
>
> We conducted the sensitivity analysis, i.e., calculating results without post-hoc zeroing, and find no statistically significant results.
> Specifically, the top-ranked answers do not change and average accuracy differences are within statistical noise (Δ < 0.01, well within our 95% CIs). This confirms that zeroed answers already had p ≤ 0.2 from empirical annotations, meaning annotators independently recognized these as incorrect.
>
> We still believe that this is an important step, even as it  encodes expert priors. Some answers are objectively wrong (e.g., billing codes that violate CMS regulations, medications contraindicated for specific conditions). Including them as non-zero (even if negligible would risk reward models for "hedging" toward incorrect options, contaminate calibration metrics with clinically dangerous responses, and conflate "ambiguous but valid" with "factually incorrect".
>
> **Weakness 2**: Stating jurisdiction-specificity not stated upfront (in the abstract)
>
> Absolutely fair point and we will add this to abstract and introduction for transparency.
>
> **Question 4**: Demographics of the 9 annotators and HBT annotator-specific parameters
>
> The MENTAT dataset is developed and overseen by a diverse group of practicing clinicians in the U.S. representing a variety of ethnic, sexual, and gender orientations, as well as different specializations in psychiatric care. All nine question designers and annotators are practitioners and M.D.s in the U.S. psychiatric care system with two or more years of clinical experience post M.D. degree.
>
> Regarding the annotator demographics being relfected in the annotator-specific parameters: With only 9 annotators across 5+ demographic dimensions, we lack statistical power for meaningful subgroup comparisons (minimum ~15-20 _per group_ needed for 95% CI separation). Our HBT model's annotator-specific parameters (as shown in Figure 14) capture individual differences in strictness/tie-breaking, but we cannot attribute these to demographic factors without larger N.
>
> **Question 1**: Release raw clinician rationales for richer qualitative studies?
>
> We support this direction with an important caveat. Most rationales were collected informally during question creation/verification (not systematically annotated), so we cannot release the majority as structured data. However, we will release anything we have. Please note that MENTAT is explicitly an evaluation-first dataset (like AIME, HumanEval), not a training corpus, being designed for discriminative rankings between models.
>
> **Question 2**: Ablations for answer-order randomization and few-shot vs. zero-shot?
>
> We already randomize answer order answer-order, for example, when creating the different fairness datasets (Evaluation prompts also randomize order across model queries) or during the annotation process randomization (see annotation interface in Figure 18, Appendix I).
>
> On few-shot, we evaluated all models in zero-shot mode to ensure fair comparison across architectures. We agree this ablation is valuable and will add few-shot results (k=3 examples from the designated 20-question few-shot set) to the rebuttal revision.
>
> **Summary**: We have conducted your requested sensitivity analysis (Weakness 1) and fairness error-type analysis (Question 3), will clarify jurisdiction scope upfront (Weakness 2), and have addressed randomization (Question 2) and annotator diversity (Question 4) transparently. Your feedback has significantly strengthened MENTAT's methodological rigor and fairness analysis. Thank you for the thorough review and for considering MENTAT for publication at ICLR!
>
> Best, The Authors

---

> > ### Comment · Reviewer_W1d3 · 2025-11-27
> >
> > Thanks for your detailed reply and experiments which cleared out my questions. Please include the expanded explanations and results in the updated version.

---

### Official Review · Reviewer_zwDT · 2025-11-10

**Soundness:** 3
**Presentation:** 3
**Contribution:** 3
**Rating:** 6
**Confidence:** 4

**Summary:**

This paper presents MENTAT, a clinician-authored dataset and evaluation pipeline for psychiatric decision-making that aims to reflect routine practice rather than exam-style fact recall. The dataset targets five domains: diagnosis, treatment, monitoring, triage, and documentation, and it is designed to study ambiguity and disagreement that arise in real care. The authors collect expert ratings and convert them into preference probabilities through a hierarchical Bradley-Terry model, which provides soft labels for ambiguous tasks. They evaluate eleven general-purpose instruction-tuned models and four mental-health-focused models across multiple-choice and free-form settings, with dedicated analyses on performance shifts caused by patient age, gender coding, and ethnicity. The results show strong performance on diagnosis and treatment, weaker results on triage and documentation, and measurable fairness gaps linked to demographic variables.

**Strengths:**

1.  The use of multiple clinicians on the same items would be valuable to show inter-rater agreement through statistics such as kappa or intra-class correlation. This would document label consistency, separate real clinical ambiguity from noise, and build confidence in the reliability of the resource. This shows label consistency, highlights true clinical ambiguity, and increases confidence in the dataset’s reliability.

2. The focus on everyday, ambiguous decision-making in psychiatry is a clear strength. MENTAT moves away from board-style recall and targets tasks that practicing psychiatrists actually perform, including triage, documentation, and longitudinal monitoring.

3. The analysis of demographic sensitivity across gender coding, age, and ethnicity is careful and informative. By constructing DG, DA, and DN variants, the authors show statistically significant accuracy differences, for example higher accuracy for non-binary-coded patients than for female-coded patients, lower accuracy for items with Asian or African American ethnicity than for other groups, and higher accuracy for the youngest age band. This investigation of fairness in clinical AI is highly relevant.

**Weaknesses:**

1. A key weakness is the free-form evaluation relies on a single semantic similarity signal. The study measures inconsistency as one minus BERTScore using a DeBERTa xlarge model fine-tuned on MNLI, then aggregates with bootstrap resampling. This narrow view can miss clinically reasonable phrasings and may not reflect calibration. Reporting multiple signals, such as ROUGE or BLEU, would give a more stable picture of free-form behavior.

2. The core dataset is small for the breadth of claims. The paper states there are 203 base questions across the five categories, and scale-ups for fairness come from demographic templating rather than new clinical scenarios. This limits statistical power for fine-grained comparisons and constrains analysis of rare but important edge cases.

3. The paper would benefit from short qualitative examples of free-form outputs alongside the preference-based labels. Concrete examples of model answers that are close to or far from expert choices would help readers understand error modes in triage, documentation, and treatment adjustments.

**Questions:**

Please see the weaknesses.

---

> ### Author Response · Authors · 2025-11-26
>
> Dear Reviewer 1 (zwDT),
>
> Thank you for your thorough review and constructive feedback. We appreciate your recognition of MENTAT's focus on everyday psychiatric decision-making and our careful demographic sensitivity analysis. Below we address your concerns:
>
> **Weakness 1**: Free-form evaluation relies on single semantic similarity signal
>
> We agree and have addressed this directly with qualitative and quantitative results. Please see "Exp 1" in the general comment for the full details. In short, we computed ROUGE-L and BLEU alongside BERTScore to provide multiple signals as you suggested. We find that the ranking of models remains consistent across BERTScore and ROUGE-L (o1 consistently best, Claude models consistently show higher inconsistency). BLEU shows uniformly high inconsistency due to its reliance on exact n-gram overlap, which is overly strict for clinical language where multiple valid phrasings exist.
>
> We manually examined free-form responses and found that low inconsistency scores consistently correspond to correct answers, while high-inconsistency responses are almost always incorrect. The rare false negatives (high-inconsistency + correct) result from valid responses with different formatting/detail levels, which we belieave to be an acceptable limitation given that minimizing false positives is more critical for clinical safety.
>
> Both results validate our choice of BERTScore as the primary metric while demonstrating robustness across complementary measures.
>
> **Weakness 3**: Adding qualitative free-form responses
>
> We fully agree and we will add them to the appendix. As noted in Weakness 1 above, we have conducted qualitative analysis and will include concrete examples showing: High-inconsistency incorrect responses (most common failure mode), Low-inconsistency correct responses (successful cases), and High-inconsistency correct responses (valid alternative phrasings).
>
> **Weakness 2**: Core dataset size
>
> We respectfully argue that size alone does not determine the dataset value, as quality and design can be equally critical. MENTAT follows successful precedents of evaluation-first benchmarks that prioritize expert curation over scale, e.g., AIME (30 samples), HumanEval (164 problems), or BIG-Bench Hard (2k questions).
>
> Our dataset is discriminative despite its size, as all reported results include bootstrap-resampled 95% confidence intervals and pass statistical significance tests, demographic parameterization (DG: 549 prompts, DA: 915 prompts, DN: 1,098 prompts) provides thousands of controlled evaluation instances for fairness analysis, and we find significant performance differences between models (e.g., o1: 0.81 overall vs. Llama3.2-3b: 0.58) and categories (diagnosis: 0.82 vs. triage: 0.44). We thoroughly check statistical significance in all experiments and applications, to be transparent about dataset size induced effects and how to interpret the results. The usability and informativeness of MENTAT is further supported by the new nuanced demographic bias analysis in "Exp 2" and the performance of the seven newer tested models in "Exp 3" in the general rebuttal comment.
>
> Still, we agree expansion is valuable and will do our best to expand MENTAT for the camera-ready version. We assumed that the current dataset successfully achieves its evaluation goals with statistical rigor and the two week review discussion period is not sufficient time to generate, annotate, and more data with our time-intensive usage of human domain experts and multipel layers to verification to ensure highest data quality.
>
> **Comment**: On inter-rater agreement statistics
>
> We already report Krippendorff's α in Appendix F in Figure 15 for all triage and documentation questions (the "ambiguous" categories). Values range from slightly negative to 0.8, confirming our design goal: capturing genuine clinical ambiguity where experts legitimately disagree, rather than measuring convergence on a single "ground truth." We intentionally did not discard low-α questions because disagreement is the phenomenon we aim to model.
>
> **Summary**:
>
> We have directly addressed your primary concerns with new multi-metric experiments (Weakness 1), provided statistical justification for dataset size (Weakness 2), and will add requested qualitative examples (Weakness 3). We believe these additions strengthen MENTAT's contribution as the first expert-curated, fairness-aware psychiatric decision-making benchmark.
>
> Thank you for considering MENTAT for publication at ICLR!
> Best,
> The Authors

---

### Author Response · Authors · 2025-11-26
**General Rebuttal Comment: New Experimental Results I**

Dear Reviewers, dear AC and SAC,

Thank you for your constructive feedback! We conducted (_Edit Dec 3rd_)~three~ four substantial experiments that strengthen our core contributions and address your concerns. Below we summarize these new results and will engage with the individual reviewers below their initial reviews.
(Please excuse the delayed response, I (the lead author) was feeling unwell the last few days and that delayed our comments.)

### TLDR/Key findings from new experiments:

* We provide additional support to the claim that MENTAT enables reliable free-form evaluation by using two additional metrics and conducting qualitative analysis of free form responses with consistency checks of the metrics
* We conduct a more nuanced failure analysis of patient demographic information-induced accuracy variation, demonstrating that statistically significant demographic biases persist across models and categories and confirm "random" discriminatory bias through qualitative analysis
* We demonstrate the value of MENTAT by adding 7 more models to the 15 we initially tested (total of 22 tested models) and find that recent open-source models close part of the gap to the tested close models
* (_Edit Dec 3rd_) We test the impact of few-shot prompting and find significant improvements in the triage category for smaller/older models but little to no effect for larger/recent models, suggesting that weaker models rely more heavily on in-context learning

## Exp 1: Additional support for using MENTAT to probe free-form model decision-making [As proposed by Reviewer 1 (zwDT)]

**Quantitative**: We calculate free-form similarity between expert and LM-generated responses with ROUGE and BLEU metrics to create a more stable picture of free from behavior and complementing our used BERTScore-based consistency metric. The order of Table 5 (Appendix F) is consistent across metrics (lowest, i.e., best, scorer remains consistent and any minor deviations between higher scoring models is within omitted 95% confidence intervals) and we observe an increase in inconsistency for BLUE, as expected due to it relying solely on n-gram overlap, increasing inconsistency. In conclusion, using ROUGE-L and BLEU as additional metrics support our initial result, while also indicating the limitation of using BLEU score as comparison metric between free-form responses. _The consistency across metrics validates MENTAT as a reliable reference for free-form evaluations._

**Diagnosis**
| Metric   | 4o   | o1   | 3.5H | 3.5S |
|----------|------|------|------|------|
| BERT     | 0.55 | 0.40 | 0.75 | 0.74 |
| ROUGE-L  | 0.44 | 0.25 | 0.70 | 0.70 |
| BLEU     | 0.95 | 0.90 | 0.98 | 0.99 |

**Treatment**

| Metric   | 4o   | o1   | 3.5H | 3.5S |
|----------|------|------|------|------|
| BERT     | 0.82 | 0.77 | 0.88 | 0.84 |
| ROUGE-L  | 0.86 | 0.82 | 0.91 | 0.87 |
| BLEU     | 0.99 | 0.99 | 1.00 | 0.99 |

**Triage**

| Metric   | 4o   | o1   | 3.5H | 3.5S |
|----------|------|------|------|------|
| BERT     | 0.75 | 0.77 | 0.79 | 0.77 |
| ROUGE-L  | 0.84 | 0.87 | 0.88 | 0.85 |
| BLEU     | 0.98 | 0.99 | 0.99 | 0.98 |

**Rebuttal revision**: We will extend Table 5 for the rebuttal revision of the paper with all uncertainties [uncertainties omitted here for markdown visibility issues].

**Qualitative**: We also study the generated free-form responses and compare them to the expert annotated ones in MENTAT to further support our free-form consistency experiments. We find that low inconsistency corresponds with a correct answer in all tested cases, while almost all high inconsistency scoring responses are in-correct. The few high-inconsistency but correct responses strained far from the original answer format (length, format (more chatbot-conversation-like), and level of detail). We argue that this is acceptable, given that a low false-positive is more crucial than a (small) false-negative rate, especially given that there are no other existing datasets that provide expert annotated clinical decision-making data.

**Rebuttal revision**: We collected a series of free-form responses across models with high/low inconsistency (according to our used BERTScore metric) and correct/incorrect verified labels to the appendix to support our qualitative analysis. We will also add the qualitative analysis to the rebuttal version.

---

> ### Author Response · Authors · 2025-11-26
> **General Rebuttal Comment: New Experimental Results II**
>
> ## Exp 2: More detailed failure analysis by demographic information [As proposed by Reviewer 2 (W1d3)]
>
> **Quantitive**: To enable a more nuanced analysis of patient demographic information impact on model performance, we list model performance across the five decision-making categories (diagnosis, treatment, …). In short, _we find statistically significant biases across all models and categories_ [All results in this subsection are reported with statistical significance.]
>
> In terms of impact of patient gender, we find that all models (top-capability models or including open-source models) perform similarly in the diagnosis category, while men and non-binary- coded patients would receive higher accuracy than female-coded patients in the monitoring, treatment, triage, and documentation categories. Between male and non-binary-coded patients, accuracy is lower for non-binary-coded patients in the triage category, clearly indicating stigma towards patient gender.
>
> Similarly, for patient age ranges, patients labeled as
> * 18-33 years old receive the highest accuracy in diagnosis, monitoring, triage categories
> * 33-49 years old best receive the highest accuracy in the treatment (but second in diagnosis and triage) category
> * 49-65 years old only receive the highest accuracy in the documentation (but second in monitoring and treatment) category
>
> In terms of patient nationality, in relative comparisons between patient demographic variables, we find that patients labeled
> * African American receive higher accuracy in the diagnosis and monitoring categories
> * Native American receive higher accuracy in the treatment and triage categories
> * White receive higher accuracy in the treatment and triage categories
> * Black receive higher accuracy in the monitoring and documentation categories
> * Asian receive higher accuracy in the diagnosis, monitoring, triage, documentation categories
> * Hispanic receive higher accuracy in the documentation, treatment, diagnosis categories
> These results indicate not a clear pattern, but a statistically significant bias across categories.
>
> **Qualitative**: Studying which question models seem to perform good or bad at depending on patient demographic and question content (within a category) does not seem to reveal a clear pattern. This finding supports the quantitative analysis above, as there seems to be no clear pattern that, e.g., models would perform better particularly good or bad in some categories for multiple minorities etc. This lack of a clear failure pattern demonstrates the need for a novel fairness-aware clinical decision-making dataset like MENTAT. As they are hard to predict from a  few qualitative samples, but can have dire consequences for individual patients, and only statistically surface across many samples.
>
> **Rebuttal revision**: We will add the patient demographic category-specific performance values for all tested models in the appendices in the final rebuttal revision with statistical uncertainties.

---

> ### Author Response · Authors · 2025-11-26
> **General Rebuttal Comment: New Experimental Results III**
>
> ## Exp 3: Testing additional models (off-the-shelf and fine-tuned for medical applications) [As proposed by Reviewer 4 (snwZ)]
>
> To the 15 language models we initially tested fifteen (eleven off-the-shelf and four (mental) health fine-tuned LMs), _we added five new off-the-shelf and two fine-tuned models to reduce the gap between open and closed models as well as use more recent specialized models_. [Statistical uncertainties omitted for markdown readability, but will be included in rebuttal revision.]
>
> | Model | Overall | Diagn. | Moni. | Treat. | Triage | Docum. |
> |-------|---------|--------|-------|--------|--------|--------|
> | **[OLD MODELS (for reference)]** | | | | | | |
> | Mentallama_7b_chat | 0.39 | 0.54 | 0.33 | 0.36 | 0.35 | 0.30 |
> | Pmc_llama_13b | 0.48 | 0.63 | 0.52 | 0.48 | 0.31 | 0.30 |
> | Mmeds-llama3_8b | 0.58 | 0.74 | 0.57 | 0.69 | 0.42 | 0.33 |
> | Llama2_7b_chat_hf | 0.39 | 0.50 | 0.33 | 0.45 | 0.35 | 0.26 |
> | llama3_2_3b_instruct | 0.58 | 0.78 | 0.55 | 0.60 | 0.46 | 0.41 |
> | llama3_1_8b_instruct | 0.65 | 0.78 | 0.60 | 0.83 | 0.50 | 0.37 |
> | Claude-3-haiku | 0.70 | 0.85 | 0.67 | 0.93 | 0.46 | 0.41 |
> | Claude-3-opus | 0.73 | 0.85 | 0.76 | 0.88 | 0.50 | 0.48 |
> | Claude-3-5-haiku | 0.71 | 0.89 | 0.60 | 0.93 | 0.51 | 0.44 |
> | Claude-3-5-sonnet | 0.77 | 0.85 | 0.83 | 0.95 | 0.54 | 0.48 |
> | Gpt4o-mini | 0.74 | 0.91 | 0.71 | 0.95 | 0.42 | 0.44 |
> | Gpt4o | 0.79 | 0.93 | 0.86 | 0.98 | 0.42 | 0.48 |
> | O1-mini | 0.75 | 0.89 | 0.86 | 0.86 | 0.50 | 0.44 |
> | O1 | 0.81 | 0.96 | 0.98 | 0.95 | 0.46 | 0.44 |
> | **[NEW MODELS]** | | | | | | |
> | gemma-3-4b-it | 0.55 | 0.62 | 0.58 | 0.45 | 0.64 | 0.48 |
> | gemma-3-12b-it | 0.66 | 0.76 | 0.58 | 0.70 | 0.60 | 0.59 |
> | Gemma-3-27b-it | 0.71 | 0.76 | 0.65 | 0.82 | 0.68 | 0.56 |
> | Qwen3-4B-Instruct | 0.66 | 0.69 | 0.53 | 0.80 | 0.60 | 0.63 |
> | Qwen3-30B-A3B-In. | 0.76 | 0.84 | 0.74 | 0.89 | 0.68 | 0.52 |
> | Internistai/base-7b | 0.64 | 0.78 | 0.47 | 0.80 | 0.60 | 0.44 |
> | Medgemma-27b-it | 0.69 | 0.73 | 0.58 | 0.80 | 0.76 | 0.56 |
> (Meditron-7b results not shown as stated in our submission)
>
> More recent models perform better on average across all categories for their parameter size. The tested closed models from OpenAI and Anthropic still outperform the newer (but smaller) open models in the categories diagnosis, treatment, and monitoring. However, they close the gap and even outperform their closed counterparts in the triage and documentation category (although not as often as the above table suggests, due to statistical uncertainties).
>
> **Rebuttal revision**: We will update the paper to include all new model results and add a version of the above table with uncertainties in the appendix.

---

> ### Author Response · Authors · 2025-12-03
> **General Rebuttal Comment: New Experimental Results IV**
>
> ## Exp 4: Testing model few-shot performances (k = 3) [As proposed by Reviewer 2 (W1d3)]
>
> We add new experiments to also see how model performances could change with few-shot prompting. For $k=3$ few shot examples from the training split (from the same question category as is being querried), we evaluate twelve open-source model performances.
>
> | Model | Overall | Diagn. | Moni. | Treat. | Triage | Docum.|
> |-------|-----|-----------|------------|-----------|--------|---------------|
> | Mentallama_7b | 0.39 | 0.54 | 0.33 | 0.36 | 0.35 | 0.30 |
> | Pmc_llama_13b | 0.56 | 0.80 | 0.47 | 0.61 | 0.52 | 0.26 |
> | Mmeds_8b | 0.60 | 0.67 | 0.65 | 0.68 | 0.68 | 0.22 |
> | Llama2_7b | 0.39 | 0.49 | 0.30 | 0.34 | 0.44 | 0.41 |
> | Llama3_2_3b | 0.54 | 0.80 | 0.42 | 0.55 | 0.64 | 0.22 |
> | Llama3_1_8b | 0.64 | 0.78 | 0.51 | 0.75 | 0.64 | 0.44 |
> | Gemma-3-4b-it | 0.55 | 0.62 | 0.58 | 0.45 | 0.64 | 0.48 |
> | Gemma-3-12b-it | 0.66 | 0.76 | 0.58 | 0.70 | 0.60 | 0.59 |
> | Gemma-3-27b-it | 0.71 | 0.76 | 0.65 | 0.82 | 0.68 | 0.56 |
> | Qwen3-4B | 0.66 | 0.69 | 0.53 | 0.80 | 0.60 | 0.63 |
> | Qwen3-30B | 0.76 | 0.84 | 0.74 | 0.89 | 0.68 | 0.52 |
> | Medgemma-27b | 0.69 | 0.73 | 0.58 | 0.80 | 0.76 | 0.56 |
>
> For smaller and mostly older open-source models (Pmc_llama_13b, Mmeds_8b, Llama2_7b, Llama3_2_3b, Llama3_1_8b), we see a significant improvement in the triage category when combined across models. Larger and more recent models do not show changes in performance, potentially indicating that the improvements are related to weaker models relying more on in-context learning to perform better in the triage category. However, besides improvements in the triage category, some models improved in the documentation category, while others declined or performance stays the same. Overall, the few-shot results are mixed and only seem to affect smaller and less recent models.
>
> **Rebuttal revision**: We added the results in appendix K.5 with statistical uncertainties.

---

### Author Response · Authors · 2025-12-01
**Rebuttal Revision Upload Comment**

Dear Reviewers, dear AC and SAC,

@Reviewers: Thank you again for your helpful feedback and seeing the value in MENTAT! Your input helped us to objectively improve our submission and we greatly appreciated the discussions, acknowledgements that our rebuttals addressed your concerns, and the resulting favorable score change(s) (at least while it was still possible).

**Summary of Rebuttal Revision Changes**

We have **uploaded a rebuttal revision with the following changes** (if applicable, all changes are marked in red in the pdf) that provide further support to our initial findings:

- Adding results of 7 new tested models (five off-the-shelf open source and two healthcare fine-tuned), increasing our total number of tested models to 22

- Besides updating all analysis results by including the new models, we also added the full individual model performance results (based on general rebuttal comment “Exp 3” below) to appendix K.3 with 95% confidence intervals to provide more detail than just Figure 3 in Section 4

- Tested two additional free-form inconsistency metrics (ROUGE-L, BLEU) and added a qualitative analysis of free-form responses (based on general rebuttal comment “Exp 1” below) in appendix K.6 in Table 10 with 95% confidence intervals to strengthen the free-form/beyond MCQA inconsistency analysis with MENTAT

- Added free-form response examples with high/low inconsistency and correct/incorrect response pairings in appendix L

- Conducted a more detailed analysis of fairness-relevant failures (based on general rebuttal comment “Exp 2” below) by looking at the impact of patient demographic variables on task-specific performance instead across task categories alone (quantitive) in new Tables 7-9, list how each model’s performance individually is impacted for each task category (quantitative) in new Figures 22-36, and add a new qualitative analysis in appendix K.5.

- _Edit Dec 3rd: We added new few-shot (k = 3) experiments with the results shown in appendix K.4 in Table 6_

Also, minor changes:

- Refined related work section and added the reviewer-proposed references

- Added additional information on diversity of our annotators  in Section 3.1

- Added new tested model details in Section 4.1 and appendix K

- Updated the analysis results in Section 4, Figure 3, Table 1 and Table 2 based on the new results mentioned above

- Adding “U.S. centric” to the abstract

What we have not done yet, but have not forgotten and will do for a potential camera-ready:

- ~Also running few-shot results with the tested models~ _Edit Dec 3rd: We added new few-shot (k = 3) experiments with the results shown in appendix K.4 in Table 6_

- Discussing known weaknesses of LLM-as-a-judge with prior work to strengthen the need for a dataset like MENTAT

Thank you so much!

---

### Meta-Review · Area_Chair_PngC · 2025-12-22

**Summary:**

The paper presents MENTAT, a clinician-authored dataset for psychiatric decision-making, aiming to reflect the complexities of day-to-day clinical practice tasks.

The paper received favorable reviews, with reviewers praising the dataset design and practicality, as well as the exhaustive presentation. Some concerns included the small-ish dataset size and positioning to existing datasets (in particular based on MCQs).

**Reviewer Concerns:**

The authors provided an extensive rebuttal. Reviewer W1d3 acknowledged the response and raised no further issues. Reviewer snwZ, who initially raised some major concerns, later acknowledged that his concerns have been addressed and has raised their score.

The only review with a negative rating is that of Reviewer 3LdC. Their concerns are related to exposition and the authors responded with clarifications during the rebuttal period. Unfortunately, Reviewer 3LdC did not respond before the closing of discussions.

**Reviewer Scores:**

Two reviewers (W1d3 and snwZ) confirmed support after rebuttals and so may have raised scores. Reviewer 3LdC did not respond, but given that most of their concerns are related to exposition, I believe they might have raised to 6. Reviewer zwDT raises the issue that "free-form evaluation relies on a single semantic similarity signal", which the authors addressed via an additional experiment presenting free-form similarity between expert and LM-generated responses. Overall, it is hard to know how Reviewer zwDT would have responded, but their initial evaluation of the paper is nonetheless positive.

---

### Decision · Program_Chairs · 2026-01-26

Accept (Poster)